# Rapid iceberg calving following removal of tightly packed pro-glacial mélange

Surui Xie [1], Timothy H. Dixon [1], David M. Holland [2,3], Denis Voytenko[1] & Irena Vaňková [2,3,4]

Iceberg calving is a major contributor to Greenland's ice mass loss. Pro-glacial mélange (a mixture of sea ice, icebergs, and snow) may be tightly packed in the long, narrow fjords that front many marine-terminating glaciers and can reduce calving by buttressing. However, data limitations have hampered a quantitative understanding. We develop a new radar-based approach to estimate time-varying elevations near the mélange-glacier interface, generating a factor of three or more improvement in elevation precision. We apply the technique to Jakobshavn Isbræ, Greenland's major outlet glacier. Over a one-month period in early summer 2016, the glacier experienced essentially no calving, and was buttressed by an unusually thick mélange wedge that increased in thickness towards the glacier front. The extent and thickness of the wedge gradually decreased, with large-scale calving starting once the mélange mass within 7 km of the glacier front had decreased by >40%.

[1] School of Geosciences, University of South Florida, Tampa, FL 33620, USA. [2] Courant Institute of Mathematical Sciences, New York University, New York, NY 10012, USA. [3] Center for Global Sea Level Change, New York University Abu Dhabi, Abu Dhabi 129188, United Arab Emirates. [4] Present address: British Antarctic Survey, Natural Environment Research Council, Cambridge CB3 0ET, UK. Correspondence and requests for materials should be addressed to S.X. (email: suruixie@mail.usf.edu)

Previous work suggests that increasing ice discharge in marginal areas at or close to the glacier front is a major process contributing to recent ice loss in Greenland[1]. Mass loss rates from marine-terminating glaciers are generally more variable than those from other glaciers because of the influence of the time-varying ice–ocean interface[2,3]. Several factors affect ice discharge rate here, including ocean water temperature[4], time-varying water levels[5,6], terminus position[7–9], and mélange extent and strength[8–11]. Better understanding of ice dynamics at the termini of marine-terminating glaciers has the potential to reduce uncertainty in total mass balance estimates of Greenland and improve projections of future sea-level change[1,2]. However, direct observations are challenging. Here, we develop a new approach to derive precise glacier and mélange surface elevation maps with high temporal resolution (2-min interval) over a broad region using a terrestrial radar interferometer (TRI)[6,12–17]. We apply this approach to the terminus of Jakobshavn Isbræ, a major Greenlandic glacier with persistent proglacial mélange.

Jakobshavn Isbræ, Greenland's fastest moving glacier, has retreated tens of kilometers in the last few decades (Fig. 1b)[4,7]. Increased subsurface melting triggered by incursion of warm ocean water has been suggested as an important contributor[4]. The glacier's terminus is now embedded in the ice sheet, with a relatively steady position, despite some seasonal advance and retreat (Fig. 1)[6,7]. However, it is unclear how stable the present terminus position will be in the longer term, since Jakobshavn Isbræ has a retrograde bed[18]. A previous study suggests that this type of glacier is conditionally stable, with stability affected by the buttressing effect of an ice-shelf[19]. Other work has shown that mélange in front of Jakobshavn Isbræ can be characterized as a weak granular ice shelf that transmits stress from the fjord back to the glacier terminus, and the buttressing force (lateral load) can be large enough to inhibit the initiation of large-scale calving events[10,11,20]. It has also been suggested that the buttressing force on the glacier terminus depends on the thickness of the mélange[20,21].

To better understand the influence of mélange on calving, we analyzed time series of digital elevation models (DEMs) derived from TRI observations of the terminus of Jakobshavn Isbræ, allowing us to monitor changes in mélange thickness. Ice flow and glacier calving were also analyzed using TRI and satellite data (below and Xie et al.[6]). Our results reveal the details of mélange behavior during a period of glacier quiescence, providing evidence that tightly-packed mélange can suppress iceberg calving.

## Results

**TRI mapped elevation time series.** We measured time-varying elevations of the terminus of Jakobshavn Isbræ with a TRI during a ~13-day campaign from 7 to 20 June 2016 (Fig. 1a). We focus on the main (southern) branch of the glacier within 10 km of the radar. Glacier ice here is significantly thicker and moves faster than ice in the northern branch. We used 2-minute intervals between scans. A new approach was developed to improve accuracy and precision of the height estimates. We used a high precision DEM (ArcticDEM, from the Polar Geospatial Center, University of Minnesota: https://www.pgc.umn.edu/data/arcticdem/) for stationary rock areas to minimize TRI errors (Methods). Errors due to systematic bias in the ground reference point and radar geometry were corrected using a priori ground elevations from ArcticDEM. Jumps in the height estimate due to phase unwrapping errors were corrected based on their relation to phase jumps. Elevation estimates are relative to a flat surface defined by fjord water using 2% of measured mélange surface heights within a polygon area that has few large icebergs (see Methods and Supplementary Fig. 1). Accuracy and precision of the DEM time series were assessed by computing root-mean-square deviations of time series for representative stationary rock points and slow-moving ice points, and comparison to predicted tides. The derived time series based on median filtering (30-minute time window) have a height uncertainty of ~20 cm at 3 km and ~70 cm at 6 km from the radar. Height uncertainty increases with the square of line-of-sight (LOS) distance to the radar (see Methods and Supplementary Figs. 1–3). This method provides more than a factor of three precision improvement compared with previous approaches[13,14], allowing resolution of new processes within the proglacial mélange, such as mélange melting, collapse, and tide-induced elevation changes.

Figure 2a shows a 1-day median DEM. The elevation time series for representative points in the mélange (c–e) and on the glacier (f) are shown in Fig. 2c–f. Except for perturbations caused by calving-like collapse events within the mélange (mélange collapses that are similar in some respects to iceberg calving, see Supplementary Movie 1), tidally induced surface elevation changes in the mélange are well-resolved. Elevation profiles and inferred thicknesses (Fig. 2b, Supplementary Figs. 4 and 5) show a distinct step-like change in surface elevation of ~10 m located 2–4 km from the glacier front. The thick mélange upstream from the elevation step-change has a wedge-like shape, thickest at the glacier front, tapering downstream. Inferred thickness of the mélange (based on TRI-derived surface elevations and assuming hydrostatic equilibrium) near the glacier terminus exceeds 400 m (Supplementary Fig. 4). During our 2-week observation period, the elevation step-change migrated toward the glacier via several calving-like collapse events, progressively removing the downstream edge of the mélange wedge (Supplementary Movies 1 and 2). By the end of our campaign, the elevation step-change in the mélange was ~2 km from the glacier front (Fig. 3j, Supplementary Movies 1 and 2).

**Tightly packed mélange wedge suppressed calving.** Satellite images show that the main trunk of the glacier did not calve for

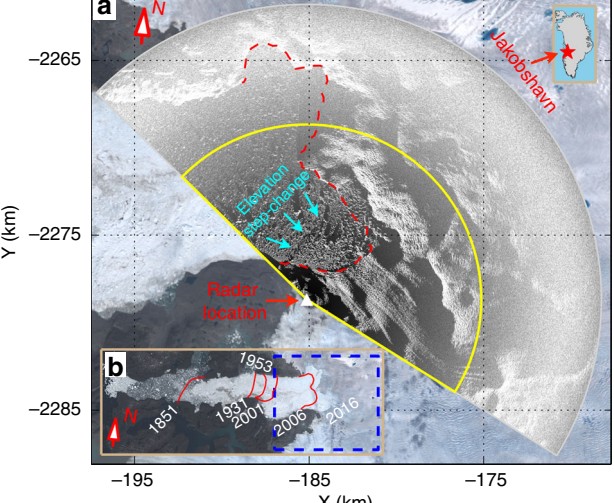

**Fig. 1** TRI scan area. **a** TRI amplitude image overlain on a Landsat-8 image (both acquired on 13 June 2016). Black areas are in radar line-of-sight (LOS) shadow. Yellow line outlines the area within 10 km of the radar, shown in Fig. 2a. Dashed red line indicates glacier front location. Cyan arrows mark the elevation step-change in the mélange. Upper right inset shows location of Jakobshavn Isbræ in Greenland. **b** Dashed blue box outlines the area presented in **a**. Red lines show calving front positions in different years, courtesy of NASA Earth Observatory. Both **a** and **b** are in polar stereographic projection (EPSG: 3413)

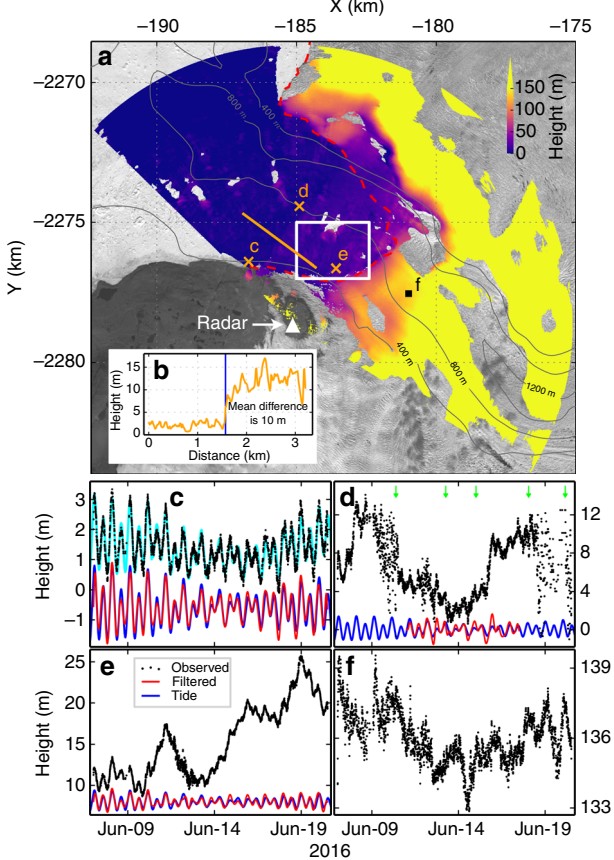

**Fig. 2** TRI-measured elevations. **a** One-day median elevation map. Dashed red line indicates glacier front location; grey contour show bed elevation[18] of the main trunk; **c–f** mark locations of elevations shown in panels below; white box outlines area shown in Fig. 5a–h; orange line marks elevation profile shown in **b**. **c–f** Black dots show observed elevations; red lines show band-pass filtered data (0.8 < frequency < 4 cycle-per-day (cpd) passed) for selected periods, shifted for clarity; blue curves show predicted tidal heights[37], shifted by the same amounts as red lines. Cyan color in **c** is the best fitting curve to data, RMS of the residuals is 22 cm (Methods and Supplementary Fig. 3). Green arrows in **d** mark major calving-like collapses (also see Supplementary Movie 1)

~17 days prior to the beginning of our TRI campaign (Fig. 3). During our TRI observations, only one small calving event occurred at the ice cliff of the main trunk, and did not cause significant motion in the nearby mélange (Fig. 3f, g; Supplementary Movie 1). In contrast, Amundson et al.[10] investigated the interactions between Jakobshavn Isbræ and its proglacial mélange using year round observations. They found that the entire lateral-width of the mélange rapidly accelerated away from the glacier even when only a small portion of the terminus fell into the mélange. We suggest that since a small calving event did not result in motion of the surrounding mélange, the mélange wedge must have been tightly packed during our observation period. Two other lines of evidence support this: first, collapses of smaller blocks at the downstream edge of the wedge caused significant downstream mélange motion but did not cause any notable motion in the mélange upstream from the elevation step-change (Supplementary Movies 1 and 2); second, during a 2015 TRI campaign at the same location and time of year, smaller calving events caused significant surrounding mélange motion (Supplementary Movie 2 in Xie et al.[6]). The 30-day period (21 May–20 June) without major calving is unusually long compared to other years at the same time of year (Fig. 4). Here, we define major

calving events as those with block size >0.25 km² and causing significant mélange motion; minor calving events are those in which visible blocks calved, but the proglacial mélange remains largely unchanged. Satellite observations show that large-scale calving events resumed within 1.5 days after the end of our campaign, causing ~9 km² ice loss from the glacier front within 8.5 days (Fig. 3j–l). Previous studies have suggested that mélange strength and iceberg calving rate (defined here as calved ice mass per day) are inversely related[10,11]. We hypothesize that calving was suppressed by the buttressing force from tightly packed proglacial mélange during a ~30-day period, from ~17 days before the beginning of our campaign until its end. Large-scale calving occurred once the mélange weakened sufficiently (i.e., the elevation step-change in the mélange migrated to <2 km from the glacier front within 1.5 days after the end of our TRI campaign). Our new DEM time series allow some aspects of this process to be quantified for the first time.

**Mélange ice mass loss**. The buttressing force of the mélange is positively correlated with sea ice and/or iceberg thickness and concentration[20–23]. Mélange ice mass (or thickness) may, therefore, be a useful proxy for mélange strength. To estimate mélange strength changes, we use total ice mass defined within a fixed proglacial Lagrangian area, and investigate changes in this mass. The close match between tidal and mélange heights (Fig. 2c–e) implies that the mélange is near hydrostatic equilibrium. A bed elevation map (contour in Fig. 2a)[18] indicates that the fjord depth here is larger than the mélange thickness. Archimedes' principle thus allows us to use our elevation time series to estimate temporal changes in mélange thickness and mass.

Several mechanisms control loss or gain of mélange ice within a given region: first, downstream advection and divergence of ice driven by glacier motion and outflow of fjord water; second, gravity-driven collapses in over-thickened mélange that enhance advection of mélange near the elevation step-changes; third, melting of the mélange driven by contact with warm air and water. Our DEM time series allow us to separate changes caused by some of these mechanisms. We calculate melt thinning (overall thickness decrease) rate based on changes of surface elevation, which are also affected by mélange divergence (details below and in Methods). To separate divergent thinning from melting, we select a box (dashed rectangle R in Fig. 5a) within 2 km of the glacier front, and track it in a Lagrangian reference frame. This selected area remained upstream from the elevation step-change until the end of TRI observations, and exhibited insignificant changes in shape and iceberg distribution pattern throughout the observation period (Fig. 5a–c and Supplementary Movie 1). Thus iceberg fragmentation should have minimal effect on melt thinning rate estimates. Each pixel within the selected area is treated as a cell with independent mobility, and the evolution of the shape and location of the selected Lagrangian area is estimated by feature-tracking[24] (Fig. 6 and Supplementary Fig. 6). Mean divergent thinning is determined by area changes of the convex hull (envelope) of all cells. Subtracting divergent thinning from the total thinning yields thinning due to melting (Methods). Because of the density difference between water and ice, TRI-measured changes in mélange surface elevation represent about one tenth of the total mélange thickness reduction. In the selected box R, TRI-derived elevations have a height uncertainty of 0.2–0.3 m, whereas the mélange thinning rate here is 0.8–1.9 m d⁻¹ (Fig. 5m), corresponding to ~0.1–0.2 m d⁻¹ in surface elevation change. Therefore, to allow sufficient signal-to-noise ratio, TRI measurements separated by multiple days are used to estimate the thinning rate. However, if a pair of TRI images is separated by too long a period of time, feature tracking

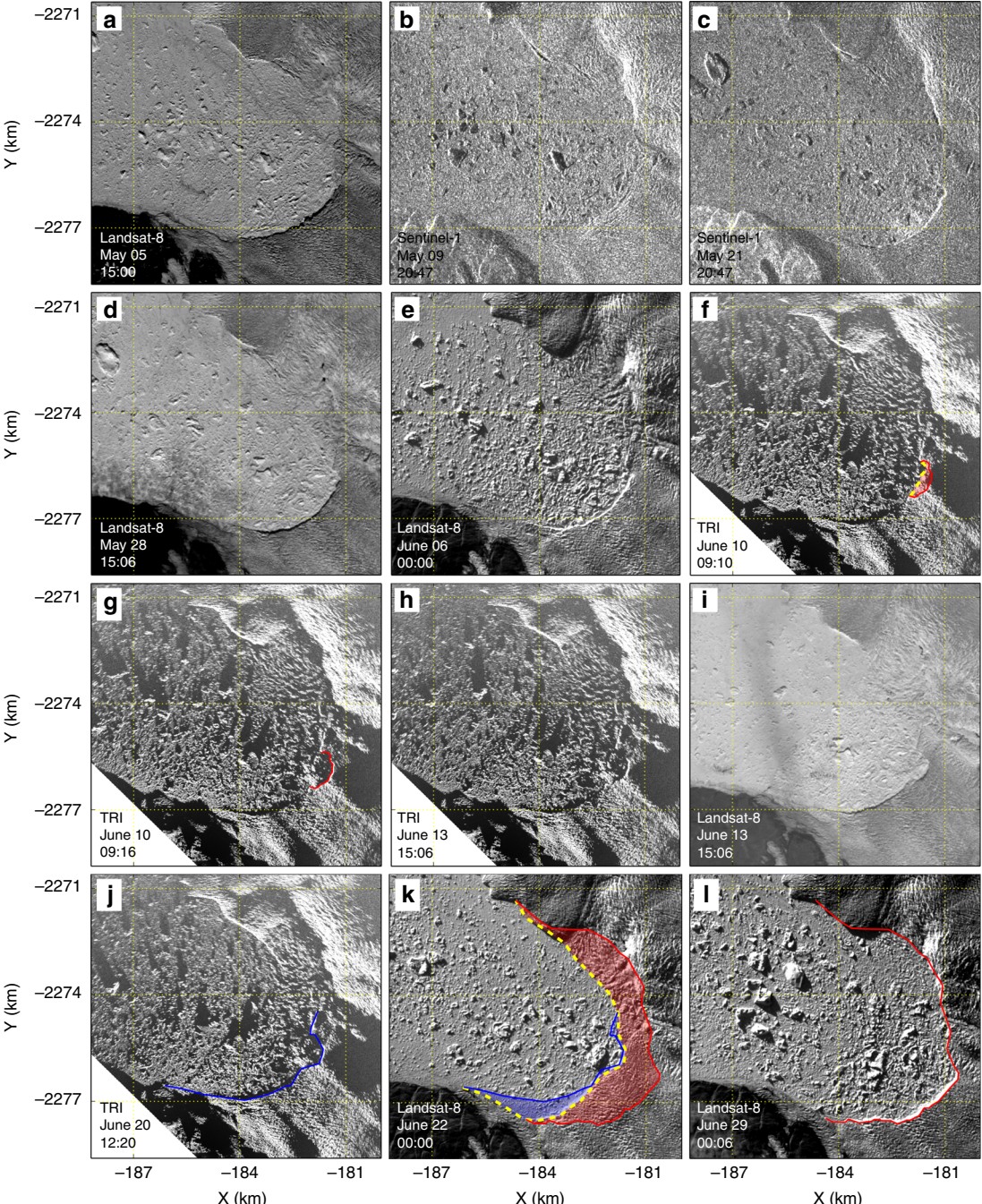

**Fig. 3** Ice loss due to glacier calving events by inspection of TRI and satellite images. For each subplot, lower left text gives platform of data source and acquisition time in 2016. In **f**, dashed yellow line shows a cliff segment which later retreated to the red lines shown in **f** and **g**, red shade shows the ice loss area due to that small calving event between **f** and **g**. Blue lines in **j** and **k** show cliff location at 12:20 20 June 2016. Red lines in **k** and **l** show location of the cliff at 00:06 29 June 2016. Dashed yellow line in **k** shows cliff location at 00:00 22 June 2016. Shaded blue and red in **k** show areas of ice loss due to calving events from digitized cliff locations. Note glacier front advanced by ~1 km between **c** and **j**

correlation decreases and the divergence uncertainty will be larger. We thus estimate average total/divergent/melt thinning rates for 6-day periods. This allows us to measure surface elevation changes that are more than a factor of 2 larger than the uncertainty in the TRI-derived DEMs, and can also give a first order estimate of the reliability of thinning rate estimates by comparison of results from different periods. Figure 5i–l show examples of elevation changes along the major axes of four selected icebergs. Figure 5m–o show the thinning rate estimates. While there is a wide range, the divergent thinning rate is always

positive, implying overall extensional motion of the mélange wedge. This may be explained by fjord geometry—the fjord widens with increasing distance from the glacier terminus. Velocity and displacement maps also show that ice motion within the mélange wedge is affected by curvature of the fjord wall (Fig. 6). Figure 5n indicates that the divergent thinning rate increased with time during the observation period, suggesting an overall increase in mélange mobility. We calculated weighted mean total and melt thinning rates and corresponding uncertainties (using one weighted standard deviation), yielding an average

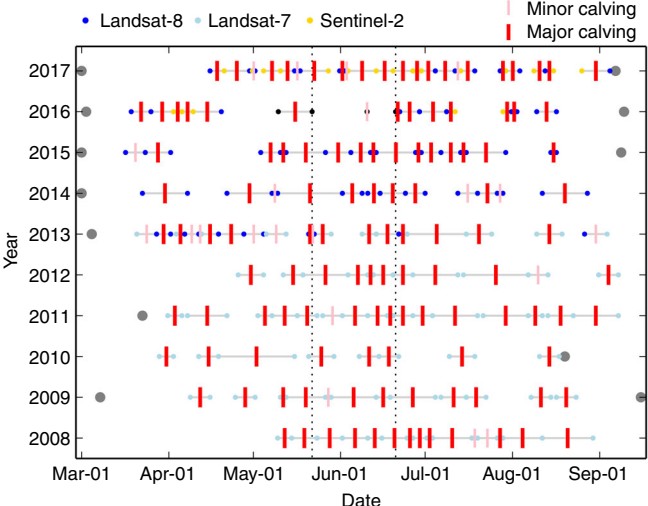

**Fig. 4** Calving events inferred from TRI and satellite images. Due to limited temporal sampling of the data, we are not able to determine the exact time of each calving event. Instead, we mark each calving event in time defined by the closest two usable images (colored dots), defined as no dense cloud coverage at the glacier front in the satellite images. Black dots at 9 and 21 May 2016 represent the acquisition times of two Sentinel-1 images. Black dots at 10 and 20 June 2016 represent acquisition times of TRI images. Two vertical dotted lines mark the 30-day period between 21 May to 20 June. For each year, the time between the grey dot on the left (first acquisition of satellite image between the study period of the year) and the useable image represent the earliest period without calving. If the image represented by the first blue or light blue dot shows no calving events compared to an image before 00:00 1 March in the corresponding year, we put the grey dot at 00:00 1 March. E.g., Landsat-8 images acquired at 14:54 26 February 2017 to until 14:54 15 April 2017 show no calving event during this period. If there is no grey dot on the left of the corresponding year (in 2008, 2010, 2012, due to cloud coverage on Landsat-7 optical images), then the first light blue dot represents the first usable image in the corresponding year. Similarly, grey dot on the right for each year indicates the last usable image that shows no calving event compare to the image shown by the last blue or light blue dot in the corresponding year. If there is no grey dot on the right of the corresponding year, it is either because the last blue or light blue dot marks the last usable image, or some calving events occurred after the last shown date in the corresponding year

total thinning rate of $1.4 \pm 0.4\,\mathrm{m\,d^{-1}}$, and an average melt thinning rate of $1.0 \pm 0.5\,\mathrm{m\,d^{-1}}$ during the TRI observation period.

Note that we assume a simple buoyancy relation in the above calculations: ice and water have constant densities and mélange is treated as an incompressible continuum (so that there is a fixed ratio between mélange thickness and above-water-height). Allowing ice density to change within a plausible range ($917^{+5}_{-30}$ $\mathrm{kg\,m^{-3}}$), and water density to vary within $1027 \pm 5\,\mathrm{kg\,m^{-3}}$ will change the mean total thinning and melt thinning rates by less than 25%. Iceberg shapes can be complex, but are impossible to define from our observations. Previous studies suggest that submerged iceberg shapes can be reasonably approximated by cylinders[25,26]. Since we only attempt to estimate an average melt thinning rate over an area, errors induced by simplifying icebergs as cylinders should be minor. Overall, these assumptions do not change the trend of our thinning rate estimates.

Our range of melt rate estimates is comparable to estimates of $0.7$–$3.9\,\mathrm{m\,d^{-1}}$ subsurface melt rates measured in summer 2008 at three nearby glaciers in the Disko Bay area[27], but is considerably larger than the $\sim 0.3\,\mathrm{m\,d^{-1}}$ estimate for an area further

downstream from Jakobshavn Isbræ between 2011–2015 from high-resolution satellite observations[26]. Our higher melt rate estimate may reflect the different locations of the study areas relative to the glacier front: The selected Lagrangian box R in our study is much closer to the glacier terminus ($<2\,\mathrm{km}$) than the study giving $\sim 0.3\,\mathrm{m\,d^{-1}}$ melt rate further downstream the fjord[26]. High subglacial freshwater discharge in summer can enhance melting near the glacier terminus by thermal plume convection, based on observations[28] and modeling[29] showing that tidewater glacier termini can have very high melt rates (exceeding $10\,\mathrm{m\,d^{-1}}$ in extreme cases), driven by subglacial freshwater discharge. Other factors, such as inter-annual variabilities in surface air temperatures or water temperatures can also cause differences in ice melt rate estimates.

Assuming that the average melt thinning rate ($1.0 \pm 0.5\,\mathrm{m\,d^{-1}}$) within the Lagrangian box R (Fig. 5a) is representative of the average melt thinning rate of mélange within 7 km of the terminus during the observation period, ice loss from melting accounts for $\sim 40 \pm 20\%$ of the observed total decrease of mélange mass in a test Lagrangian area (dashed polygon P in Fig. 7a–c). The rest can be attributed to calving-like collapse events or divergent motion within the mélange that helps to advect ice away. The two processes are not independent. Melting can break the gravity-buoyancy equilibrium, causing calving-like collapses within the mélange. Collapse events can change fjord water stratification and circulation, allowing greater ice mobility.

Figure 7a–c shows selected mélange elevation changes through part of the observation period. Supplementary Movie 2 shows changes during the entire observation period. The elevation step-change in the mélange migrates toward the glacier front (Fig. 7a–d) with significant mélange ice removed by calving-like collapses. Migration of the elevation step-change is not linear: It moves downstream between two calving-like collapses, but jumps upstream during each calving-like collapse (Supplementary Fig. 5c). The TRI data can be used to calculate the change of total ice mass within the test Lagrangian area (dashed polygon P in Fig. 7a–c). By inspecting changes in the TRI and available Landsat-8 and Sentinel-1/2 images, we can also estimate the glacier's calving rate in daily increments over $\sim 40$ days bracketing the TRI campaign (red line in Fig. 7e). There is essentially no calving from 21 May to 20 June 2016, except for a minor event on 10 June that did not significantly affect nearby mélange (Fig. 3). The coincidence of a thick, tightly packed pro-glacial mélange wedge and the absence of major calving events during an unusually long period (21 May–20 June; Fig. 4) suggests that tightly packed mélange suppressed calving. Subsequently, mélange melting and removal by calving-like collapses (totaling $1.0 \pm 0.1\,\mathrm{Gt}$ between 7 and 20 June 2016) reduced the buttressing force, eventually leading to major calving by June 21 or 22 (large-scale calving events occurred on these days, based on inspection of TRI and satellite images, see Fig. 3). Assuming that the average ice thickness at the glacier front is $800\,\mathrm{m}$, then the total ice mass calved over the 8.5 day period from June 20–29, 2016 would be $6.7 \pm 0.8\,\mathrm{Gt}$, nearly 3% of Greenland's average annual mass loss between 2003 and 2014[30].

**Decrease of mélange buttressing force.** A study of Store Gletscher used a longitudinal coupling model to explain speed increase at the glacier front associated with clearing of the mélange, suggesting an inverse relation between ice speed and buttressing force[31]. In our study, ice speed at the glacier front did not show a significant response to mélange changes during the TRI observation period (Fig. 8b, c1, c2 and Supplementary Fig. 7). However, mélange immediately upstream from the elevation step-change did show rapid increases in speed in response to

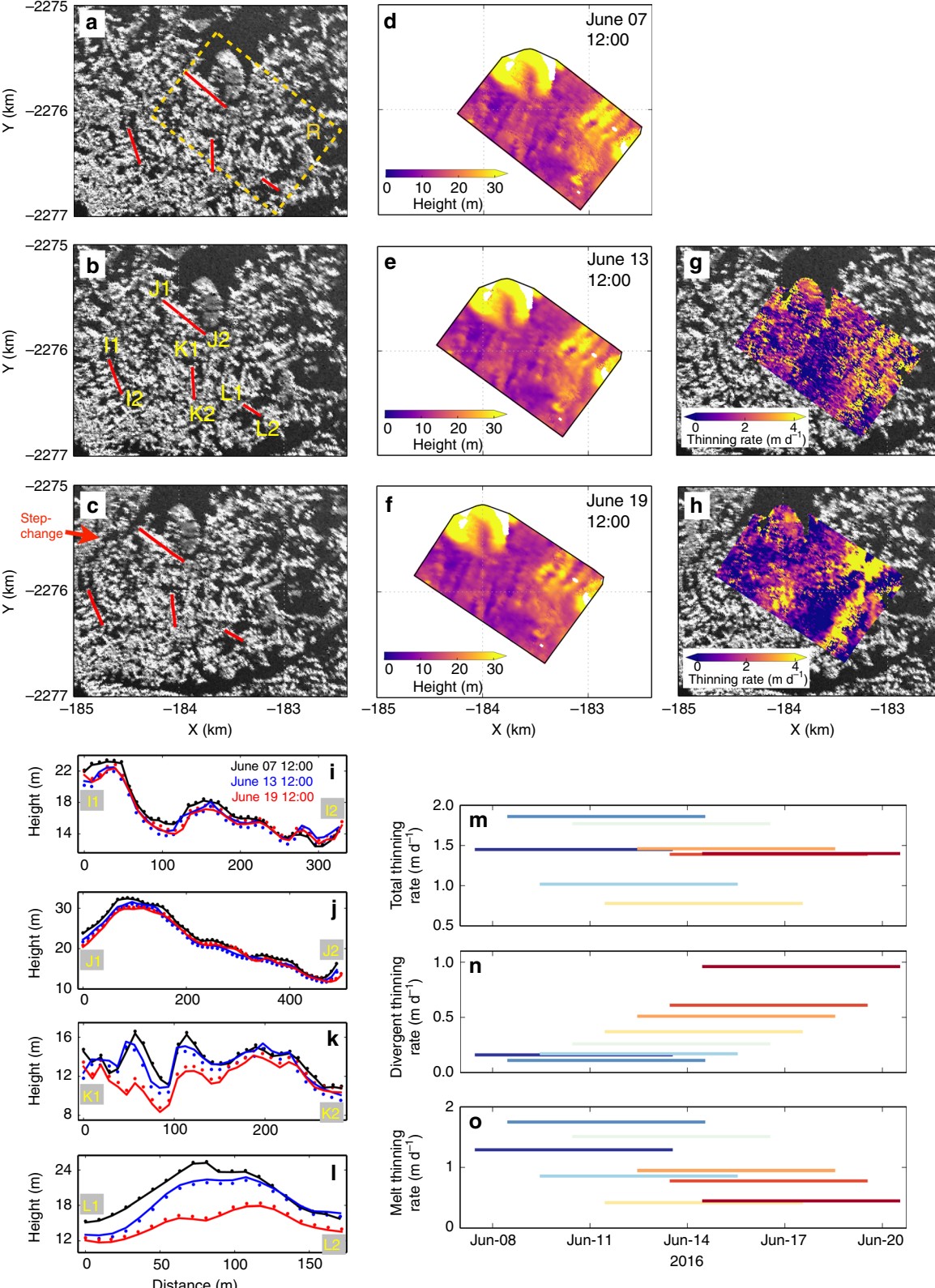

**Fig. 5** Mélange thinning. **a–c** TRI amplitude images, corresponding to times shown in **d–f**. In **a**, dashed rectangle R outlines an initial box used to estimate mélange thinning, convex hulls of the evolving box are shown in **d–f**; red lines are selected profiles to show examples of mélange thinning in **i–l**, their evolving shapes and locations are shown in **b** and **c**. **d–f** Tidal height subtracted surface elevation of mélange within the selected box, shapes and locations in **e** and **f** are estimated based on feature tracking. **g** Thinning rate from **d** to **e**. **h** Thinning rate from **e** to **f**. **i–l** Surface elevation profiles corresponding to I1–I2, J1–J2, K1–K2, and L1–L2 labeled in **b**. Black, blue, and red lines correspond to times labeled in **i**. Dots show observed elevations, lines have tidal height subtracted. Due to >30 m d$^{-1}$ moving speed of ice in the study area, an Eulerian reference frame is not suitable for mélange thinning estimate, therefore a Lagrangian reference frame is used. **m–o** Inferred mean thinning rates from measurements separated by 6 days, color from dark blue to dark red corresponding to the first and last image pairs

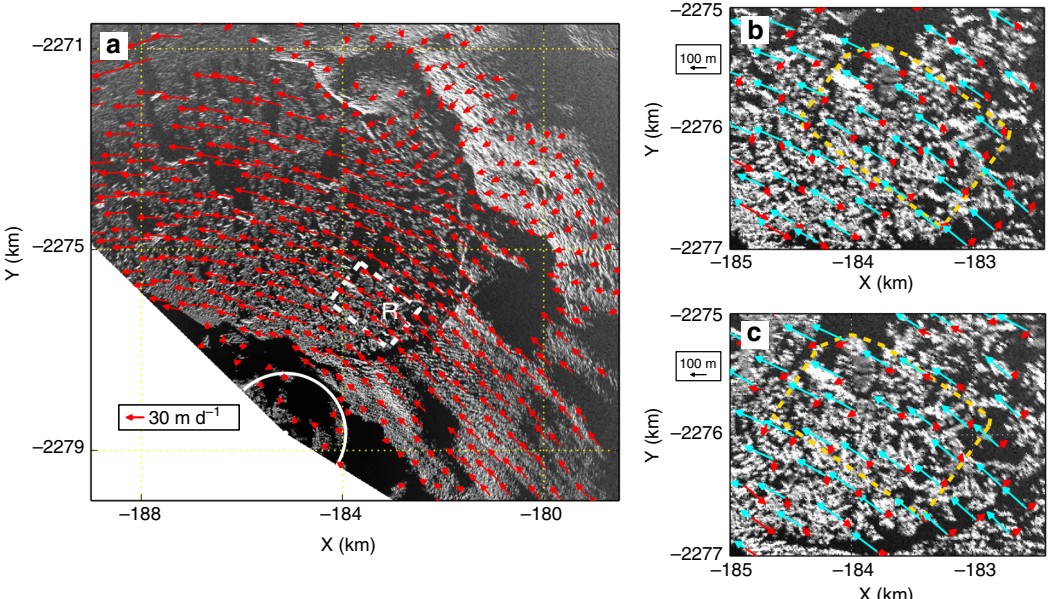

**Fig. 6** Feature tracking of TRI amplitude image pairs using the Open Source Computer Vision Library (https://opencv.org/). **a** Feature tracking velocities of a TRI amplitude image pair acquired at 12:00 7 June 2016 and 12:00 8 June 2016 (background image). Vectors within the white arc area show velocities at stationary or slow-moving points. We use their RMS as a measure of the uncertainty for feature tracking: 0.24 and 0.04 m d$^{-1}$ for two orthogonal directions. Dashed rectangle R marks the area used for mélange thinning estimates in Fig. 5. **b** Feature tracking displacements from 12:00 7 June 2016 to 12:00 13 June 2016 (background image). Cyan arrows show displacements between two TRI scans, red arrows show displacements after subtracting the median displacement within the range of the map. Dashed box corresponds to the convex hull in Fig. 5e. **c** Feature tracking displacements from 12:00 13 June 2016 to 12:00 19 June 2016 (background image). Cyan and red arrows denote the same as **b**, dashed box corresponds to the convex hull in Fig. 5f

calving-like collapses within the mélange (Fig. 8e1, e2). While the driving force for long-term mélange motion did not change significantly, the coincidence of calving-like collapses in the mélange and increases in mélange speed near the elevation step-change likely reflects reduction in buttressing force at that downstream location, presumably caused by removal of thick mélange ice. In contrast, mélange >1.5 km upstream from the elevation step-change did not show significant speed perturbations following these calving-like collapse events (Fig. 8d1, d2). We surmise that the thick mélange upstream from the elevation step-change is tightly packed, behaving essentially as an ice shelf (albeit a weak one), as suggested by previous work[10,20]. In this way, calving-like collapses within the mélange did not initially change the buttressing force at the glacier front significantly, perhaps reflecting rapid decay in stress transmission with distance.

Further lines of evidence support the hypothesis that the initial mélange collapse events did not affect buttressing of the glacier front, but did affect nearby mélange upstream from the elevation step-change. In strain rate maps (a more sensitive indicator of buttressing force change) calculated along the radar LOS (Methods), new extensional fissures appeared upstream from the elevation step-change immediately after each calving-like collapse, corresponding to the subsequent elevation step-change that would form during the next collapse event (Fig. 9 and Supplementary Fig. 8). If the buttressing force at newly formed elevation step-changes decreased significantly with each calving-like collapse event, the shear stress at the two sides of the fjord constraining the thick mélange wedge must presumably have increased, in order to prevent rapid collapse of the remaining mélange wedge. Further calving-like collapse events within the mélange occurred between the end of our TRI observations and the first available Landsat-8 image (within 1.5 days), moving the elevation step-change closer to the glacier front (the fissure marked by the cyan arrow in Fig. 9d likely failed in the next

calving-like collapse event after the TRI observation period). At some point the increased shear stress at the margins of the mélange wedge exceeds the yield stress, leading to collapse of the remaining wedge, and removal (or significant reduction) of the buttressing force on the glacier front. At this point, major calving events can occur at the glacier terminus. Note that, within uncertainties, neither the LOS nor the horizontal glacier speeds and longitudinal strain rates near the calving front changed significantly either before or after the major calving events (Fig. 8b, Supplementary Figs. 7 and 8). This may reflect reconfiguration of the terminus as the glacier front retreats, and changes in floatation status[5]. However, by the end of the TRI observation period, the glacier front close to the coast (dashed cyan outlined area of Fig. 9d) had more high strain rate zones than early in the TRI observations (Fig. 9a–c, Supplementary Figs. 7 and 8). This region of increased strain rate is also adjacent to the area that calved within 1.5 days after the end of the TRI observations (shaded blue in Fig. 3k). Combined with the progressive formation of newly formed fissures in the mélange, this observation supports an overall decrease of mélange buttressing force during the TRI observation period.

We have proposed that loss of the wedge-like mélange immediately in front of the glacier contributes to renewed calving. We now attempt to quantify this effect in terms of changes in buttressing force, using two approaches[20,21,31,32] (see Methods).

First, assuming that the mélange acts as a weak granular material[20], buttressing from the downstream thin mélange is a resistive force that prevents or limits calving-like collapses within the upstream mélange wedge. This buttressing force will decrease to a low value (and possibly zero) immediately after each collapse and then increase until the next collapse. This is supported by our data: ice surface velocity immediately upstream from the elevation step-change increases stepwise after calving-like

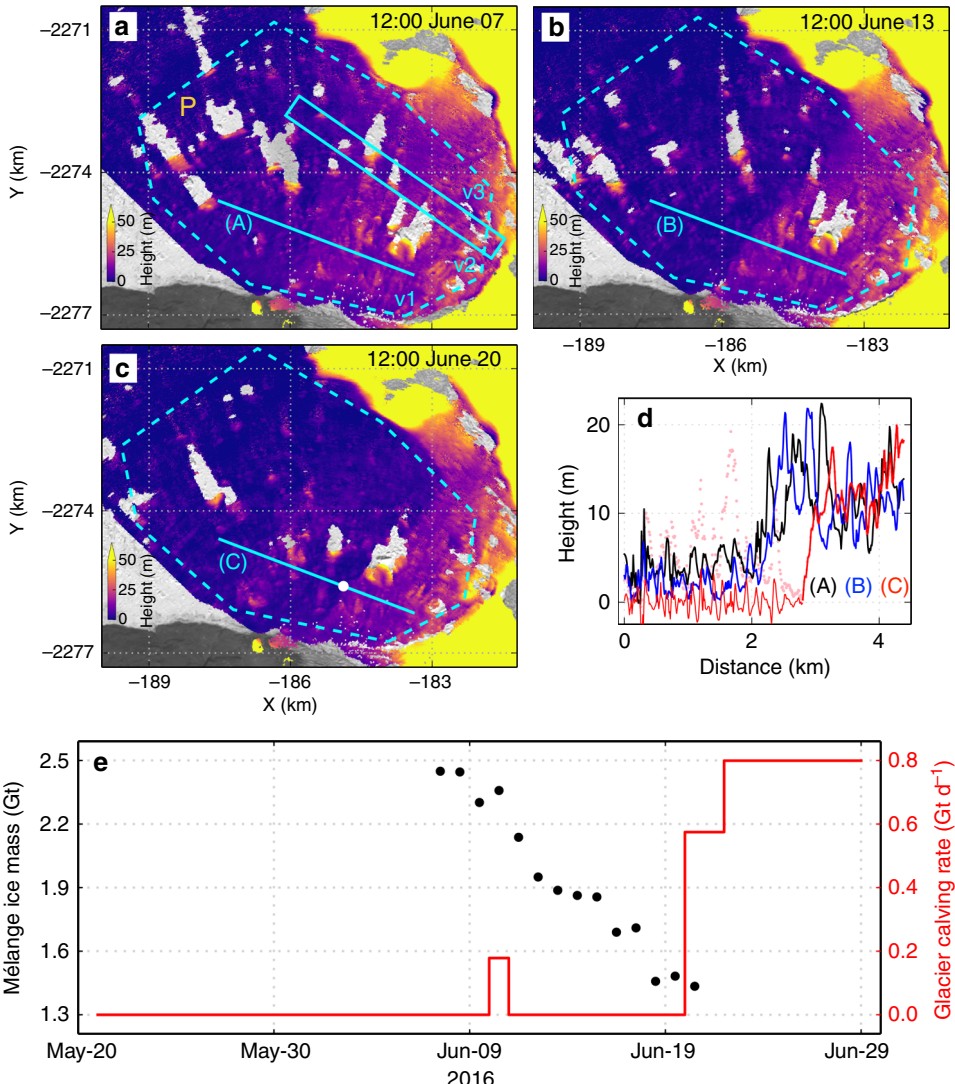

**Fig. 7** Mélange ice loss and glacier calving rate. **a–c** Surface elevation maps at selected times. Dashed polygon P outlines a test area used to calculate ice loss, moving with the three vertices v1–v3 adjacent to the glacier (marked in (**a**)). Cyan rectangle in (**a**) marks an area whose elevations are shown in Fig. 10a. **d** Elevations of an Eulerian profile marked by cyan lines in (**a**–**c**), large icebergs downstream from the white dot of (**c**) (shown with pink dots) were filtered for clarity. **e** Black dots show inferred ice mass within the test area; red line shows calving rate in daily increments

collapses (Fig. 8e1); new extensional fissures formed in the mélange wedge immediately after each calving-like collapse (Fig. 9); DEMs show that ice thickness reaches a minimum immediately downstream from the elevation step-change (Fig. 10a). Under this condition, buttressing force from mélange downstream from the elevation step-change during periods of mélange quiescence approximates the buttressing force reduction acting on the remaining thick mélange wedge immediately after each calving-like collapse. This also represents reduction in buttressing force on the glacier front immediately before major calving on 20 or 21 June 2016, assuming the thick mélange wedge had retreated to a minimum by then. Applying the model of Burton et al.[20] and simplifying the mélange as a cuboid (Methods and Supplementary Fig. 9), we estimate a buttressing force per unit lateral-width of $1.1 \times 10^7$ N m$^{-1}$. This is a minimum estimate of buttressing force decrease at the glacier front by the beginning of major calving on 20 or 21 June 2016, as it does not account for the contribution of the mélange wedge.

Second, we assume that the mélange buttressing force is proportional to mélange thickness[21]. Using the mélange buttressing stress derived from the study of Store Gletscher[31] as

representative of Jakobshavn Isbræ, and taking the average mélange thickness of the test area (Polygon P shown in Fig. 7a) as an estimate of effective mélange buttressing thickness, the decrease of buttressing force per meter of lateral-width during the TRI observation period is $\sim 0.9$–$1.8 \times 10^7$ N m$^{-1}$, equal to $\sim 11$–$22$ kPa pressure change on the entire glacier front assuming it has a thickness of 800 m. This buttressing force decrease will be even larger, reaching $\sim 2.1$–$4.3 \times 10^7$ N m$^{-1}$ (equal to a $\sim 27$–$54$ kPa pressure change upon the entire glacier front assuming it has a thickness of 800 m) by the beginning of major calving events when further calving-like collapses moved the remaining mélange wedge away within 1.5 days after the end of TRI observation period (Methods).

Amundson and Burton[33] modeled winter mélange at Jakobshavn Isbræ as a quasi-static granular material and found a buttressing force of the same magnitude as estimated above. Previous work has suggested that a back-force from the mélange of order $\sim 1.0 \times 10^7$ N m$^{-1}$ is sufficient to decelerate an already overturning iceberg or prevent an iceberg from overturning in the first place[10,20]. A finite element model suggested that back-force of this magnitude is sufficient to reduce fracture propagation near

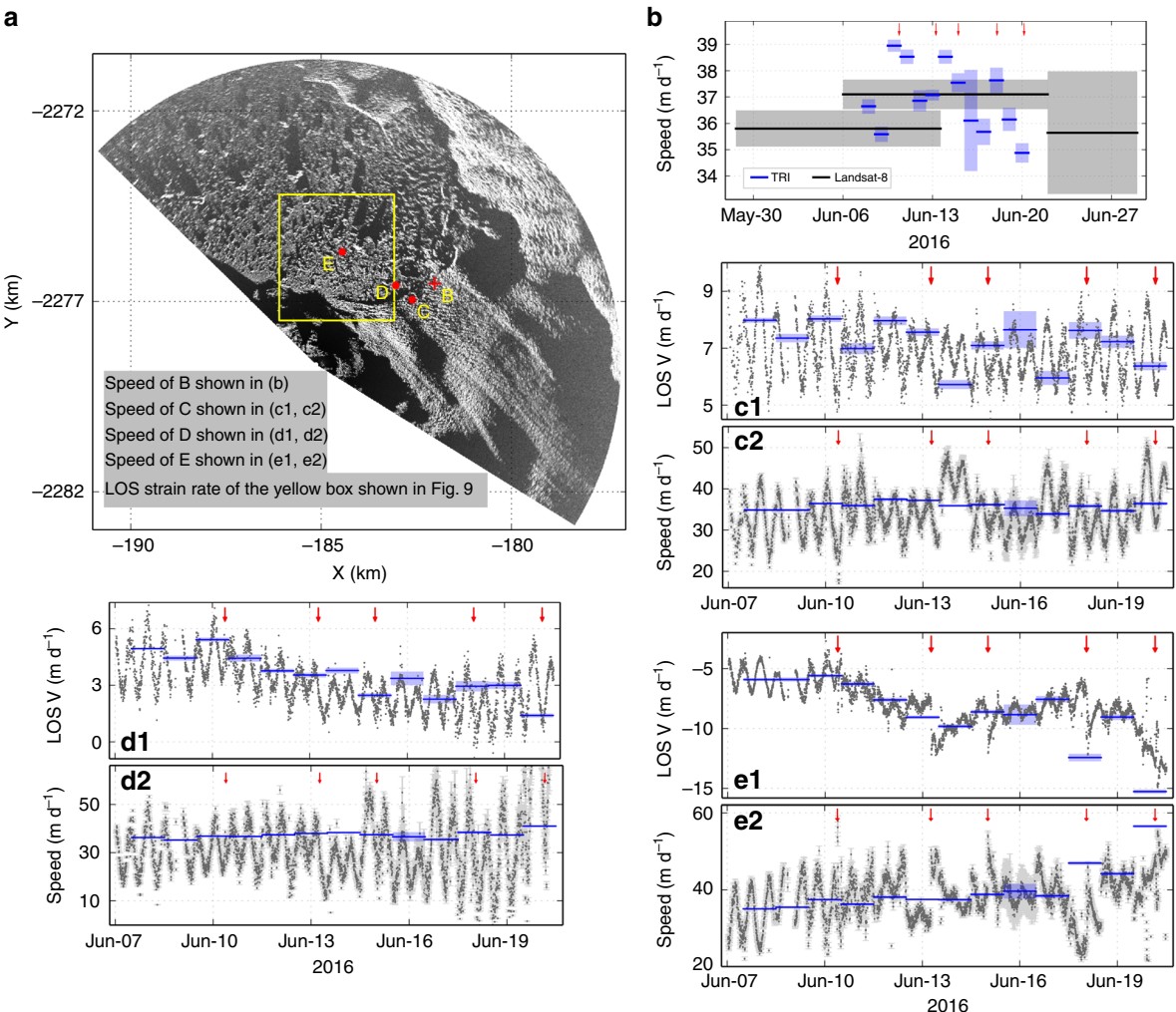

**Fig. 8** Speed responses to calving-like collapses. **a** B marks a Lagrangian point that is ~200 m to the glacier front. **C–E** represent three Eulerian points: **C** is on the glacier and within 600 m of the front throughout the TRI observations; **D** is in the mélange adjacent to the glacier and by the end of TRI campaign it was ~100 m from the glacier front; **E** is another mélange point and by the end of TRI campaign, it was ~300 m upstream from the elevation step-change in the mélange. **b–e2** Blue lines show (LOS-projected) feature tracking speeds from TRI pairs, transparent blue marks uncertainty; black lines in **b** show feature tracking speeds from Landsat-8 pairs, grey marks uncertainty. Red arrows mark major calving-like collapse events within the mélange, some of these events can last >3 h and red arrows mark times when the TRI images show most significant capsizing motion. **c1, d1, e1** Grey dots show TRI-derived LOS velocities from interferometry[6] (positive is defined as ice moving towards the radar, negative is away from the radar). **c2, d2, e2** Grey dots with light grey error bars are inverse-projected speeds based on LOS velocities and observation geometry, assuming that the angle between radar LOS and ice velocity is the same as estimated from feature tracking of an image pair separated by a day. Note instantaneous speed increase (decrease of positive LOS velocity) in **e1** after each major calving-like collapse event. Also note that a jump at 12:00 13 June 2016 in **c2** does not correspond to any calving-like collapses (the closest calving-like event is ~6 h before) and is likely caused by error in flow direction estimate

the upper surface of the glacier front by reducing tensile stress here[21]. Based on the >400 m maximum inferred thickness of the mélange wedge during our observation period, when tightly packed mélange extended down a significant fraction of the glacier front, we hypothesize that a thick mélange wedge can also reduce growth and propagation of basal fractures (Fig. 10a, c). Elevation data at the same location and the same time of year in 2015[6,17] illustrate the contrasting scenario with a thin mélange (Fig. 10b, d). Due to similarities in the speed and strain rate responses to calving-like events between the mélange wedge and glacier, we cannot distinguish whether part of the glacier front was actually detached ice blocks whose rotations were inhibited by the presence of thick mélange.

Our new observations yield direct support for the hypothesis that tightly packed mélange can suppress iceberg calving. While this is consistent with previous research[8–11,20,21,33], our

observations and analysis provide important new insights. In particular, these new data provide a quantitative framework to map tidal-timescale or shorter timescale elevation variations of pro-glacial mélange and their influence on calving across the entire glacier front. To our knowledge, this is the first quantitative study of mélange changes at daily and sub-daily timescale, and the first observation of a step-like boundary within the mélange, separating low elevation, loosely packed downstream mélange from a wedge of more tightly packed mélange near the glacier front. Past estimates of mélange thickness used in modeling either relied on limited data (characterised by low-spatial resolution and/or long revisit times)[32–34] or assumed a uniform thickness for the mélange[20,21]. Our observations clearly show a distinct thickness change in the mélange within a few kilometers of the glacier front during periods of suppressed calving. The TRI technique and our approach can be applied to other tidewater

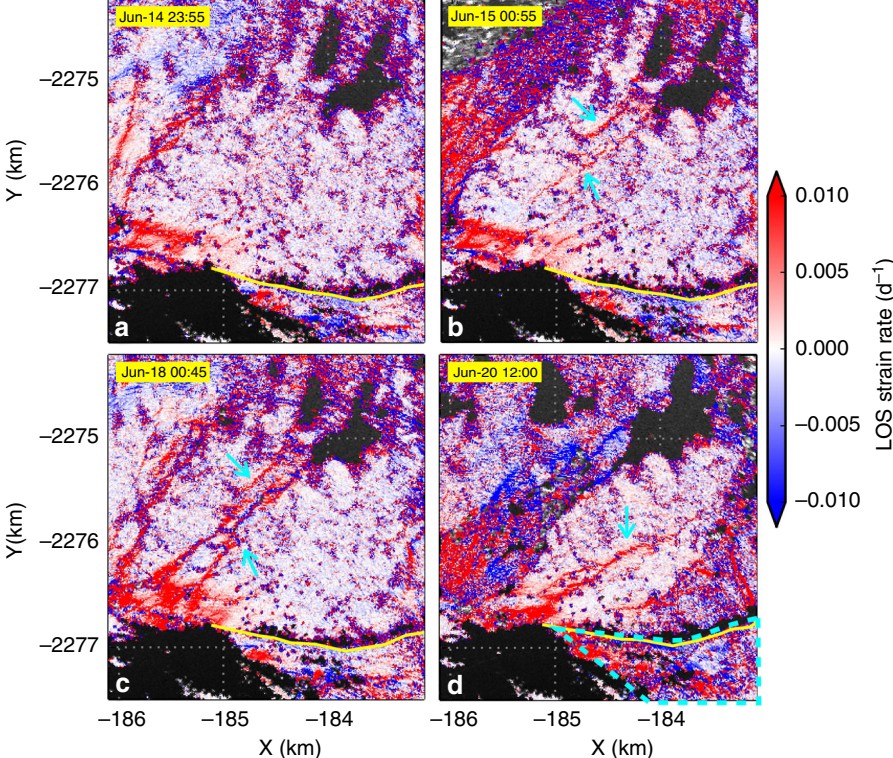

**Fig. 9** LOS strain rate responses to calving-like collapses. Strain rates are calculated along line-of-sight directions, based on 30-min median LOS velocity maps derived from interferograms of adjacent radar measurements separated by 2 min. Cyan arrows mark newly formed fissures after calving-like collapse events. **a**, **b** are at 15 min before and after a major calving-like collapse on 15 June. Fissures in **b** evolved wider to **c** and failed on 20 June. The arrow in **d** marks a strain rate fissure by the end of TRI observation period. Dashed cyan in **d** outlines an area where new high strain rate zones appeared by the end of TRI observations, which is close to the calved area within 1.5 days after the TRI observation period (shaded in blue in Fig. 3k). Map area shown in this figure is outlined by the yellow box in Fig. 8a. Yellow lines mark the mélange-glacier boundary

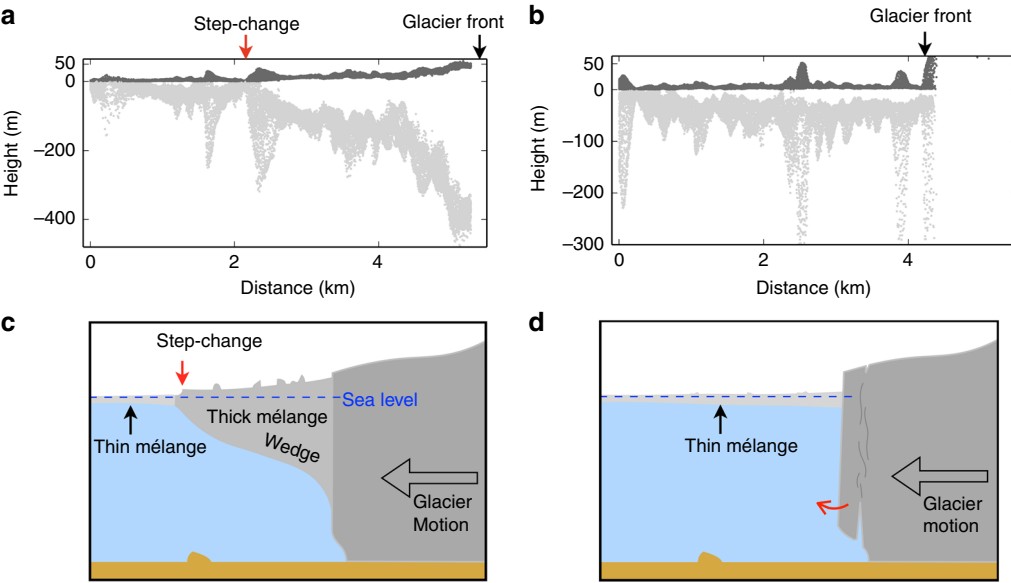

**Fig. 10** Mélange wedge and its impact on glacier calving. **a**, **b** Dark and light grey correspond to surface (measured) and bottom (inferred) heights in the outlined rectangle in Fig. 7a, distance is along profile from downstream. **a** is from this study (surface slope of the wedge is typically 0.2–2°) and **b** is from a 2015 campaign[6,17] when the mélange is relatively thin. **c**, **d** Schematic configurations along flow section of the glacier terminus with and without the thick mélange. The thick, tapered mélange in a constrained channel buttresses the glacier front and reduces fracture propagation and calving

glacier systems. Given that iceberg calving at large outlet glaciers is a major mass loss process in Greenland, and mélange can buttress the calving front and reduce calving, accurate observation and modeling of the influence of ephemeral to perennial proglacial mélange may contribute to improved understanding of dynamic ice sheet changes.

## Methods

**DEM generation and uncertainty assessment.** The TRI we use has one transmitting and two receiving antennas[6,12–17]. To generate DEMs, data are collected by both receiving antennas to form interferograms. Assuming the interferometer is vertical, unwrapped phases can be converted to elevations[13] using

$$z = \frac{\lambda}{2\pi} \frac{R}{B} \phi + \frac{B}{2} - \left(\frac{\lambda}{2\pi}\right)^2 \frac{\phi^2}{2B}, \quad (1)$$

where $z$ represents surface topography (height between radar and the study point), $\lambda$ is the radar wavelength (1.74 cm), $R$ is the range from the radar to the study point, $B$ is the baseline length (distance between receiving antennas), and $\phi$ is the phase. Depending on the application, a typical value for $B$ is ~25 cm[13], representing a compromise between precision in the phase difference measurement (related to the DEM precision, where larger $B$ values are preferred) and the ability to avoid phase breaks (phase unwrapping error, where smaller $B$ values are preferred). In this study, we chose a relatively large $B$ value (60 cm) and developed new approaches to minimize phase unwrapping error and other sources of error in the derived DEMs, discussed below.

Two steps are needed to estimate a DEM from an unwrapped phase map: first, to estimate the offset between unwrapped phase at the elevation reference point and calculated phase based on Eq. (1); second, to estimate heights at points of interest based on the unwrapped phases plus the phase offset from the first step. In the first step, elevation of the radar was measured with a single frequency GPS, and the resulting uncertainty can exceed 10 m. In addition, high-precision ground control points were not available. We, therefore, used radar and reference elevations estimated from the ArcticDEM data, provided by the Polar Geospatial Center at the University of Minnesota (https://www.pgc.umn.edu/data/arcticdem/). The absolute accuracy of ArcticDEM in this area has not been verified, however, this does not significantly affect the precision of our final estimate of TRI-derived DEMs, also discussed below.

For each unwrapped TRI phase map, when using the height difference of $z$ (including uncertainty) between the radar and reference point to estimate the phase at the reference position, we could solve a quadratic equation of one unknown ($\phi$) based on Eq. (1). However, due to uncertainties in the ArcticDEM, baseline error, and imperfect vertical mounting of the interferometer, the phase estimate at the reference point may have an error. Also, baseline error and the tilt of the interferometer propagate into the Eq. (1) used to calculate the elevation maps. Ignoring terms that can cause error at very low levels (<1 cm), the uncertainty in the height estimate is[13]

$$\sigma_z = \frac{\lambda}{2\pi} \frac{R}{B} (\sigma_{\phi 0} + \sigma_\phi) - \frac{\lambda}{2\pi} \frac{R}{B^2} \sigma_B + R\sin(\theta)\sigma_\theta, \quad (2)$$

where $\sigma_{\phi 0}$ is phase error due to errors in the reference heights and instrumental geometry, $\sigma_\phi$ is random noise in the phase measurement; $\sigma_B$ is baseline error; and $\sigma_\theta$ is the error caused by assuming the vertical axis of the three antennas is perfectly vertical, with an angle of $\theta$ from LOS direction. Note that except for random noise in the phase measurement, the other error sources are systematic, in the sense that they will cause similar errors that propagate across the entire elevation map. Ideally, with accurate knowledge of multiple ground control points, $\sigma_{\phi 0}$, $\sigma_B$, and $\sigma_\theta$ in Eq. (2) can be explicitly solved and corrected[13]. However, high precision ground control points are not available in our study area. Typical error in the baseline determination is at the 0.1 cm level, and typical error in the tilt angle of antennas is at the 0.1° level[13]. These two error sources typically cause smaller errors compared to the phase error, therefore the dominant error is linearly dependent on $R$. Thus we use a linear model to correct the majority of errors based on Eq. (2). Remaining error (e.g., the last term in Eq. (2) which is not linearly proportional to distance) will be discussed later.

Supplementary Fig. 1b shows the difference between the elevation estimates and the ArcticDEM for points near the radar (small white triangle and square, marked as 1 in Supplementary Fig. 1a). An obvious trend can be seen, indicating possible errors in the ArcticDEM, the baseline, or front-back tilt of the rack structure that supports the antennas. We excluded side-to-side tilt because it would cause a conical surface on the elevation map, which is not seen. Also, the antenna rack was mounted on stable rock, the antennas were bubble-leveled, and the system was protected from wind by a radome. We thus use a simple 1-D model based on the best fitting line of dH vs. slant range (red line in Supplementary Fig. 1b) to correct errors related to reference elevations and instrument geometry. Note that this assumes no systematic spatial bias in the ArcticDEM. Ideally, more evenly distributed points with known a priori elevations should be used as references for this correction. However in our case, only limited stationary areas were in the radar view.

Grey dots in Supplementary Fig. 1c represent the heights (1-day median) of all points within a test area (white box 2 in Supplementary Fig. 1a) in the mélange, and show an obvious trend with distance. Assuming mélange ice is in gravity-buoyancy equilibrium, no obvious trend should exist. This trend is possibly due to incomplete elimination of errors in the first-stage correction described above, because we simply used a linear correction model determined by limited near-field points. Two error sources can bias elevation estimate in the mélange: first, errors in elevations of the limited near-field points used in Supplementary Fig. 1b; second, incomplete removal of error due to imperfect vertical mounting of antennas using a linear term. The first type of error can be eliminated if more evenly distributed points with accurate elevations are available. For the second type of error, we note that our critical observation area is >2 km from the radar, where surface elevation varies at a level of 10 s of meters, the nonlinearity will only cause errors at the cm level. However, the data used to fit the linear model in Eq. (1) are in the range 0.4–2.5 km from the radar, with up to ~200 m height difference. A linear model based on these data can significantly bias the last term in Eq. (1), and then propagate through the entire DEM. We also note that elevations in the ArcticDEM are referenced to the WGS84 ellipsoid, leading to an offset between the height datum of TRI-derived DEMs and local sea level.

Both of the above two types of errors can be removed or minimized using measured heights within a relatively flat area in the mélange. We fit a plane to measured heights within box 2 (Supplementary Fig. 1a, downstream from the step-change in the mélange) that is closely parallel to local sea level (assuming mélange ice in the box maintains a flat surface defined by fjord water). We use an iterative least squares method to fit heights whose detrended values fall between 2 and 80% (marked by solid and dashed red lines in Supplementary Fig. 1c), so that the weights of random noise and measurements from large icebergs are reduced. The best fitting plane (marked by the blue line in Supplementary Fig. 1c) is then used to correct all remaining errors and offset. We use 2% of detrended heights from a 1-day median to define the mean local sea level (red line in Supplementary Fig. 1d), yielding DEMs relative to local sea level. This is a conservative criterion since it assumes 2% of measured heights are underestimated—if no error exists, the lowest height should define an upper bound of local sea level. Note that the mean sea level defined by this method may have an offset to the actual mean local seal level, however this offset will be constant for all measurements, and hence will not adversely affect our analysis of elevation change through time. Similarly, even if the box we choose to define the plane is not identically parallel to local sea level, it will not cause a time-varying signal in subsequent analysis. We also note that the correction is based on a best fitting model for areas with similar distance to the radar as box 2, but may induce a small systematic offset for areas much further or much closer than this distance. For the main observation area in the proglacial mélange that is the focus of this study, including changes through time, the model works well.

After applying these corrections, no distance-dependent trend was found in the DEMs. However, a small fraction of elevations still have large offsets (see dots marked by red arrows in Supplementary Fig. 1e). These are caused by phase unwrapping errors when incorrect numbers of phase cycles (phase jumps) were used in connecting different areas on a phase map. This can be corrected by adding an integer number of phase cycle to the unwrapped phase used in Eq. (1). For the elevation time series in Supplementary Fig. 1e, red dots are elevations fixed by adding one phase cycle to incorrectly unwrapped phases, consistent with the majority of elevations at that location. We note that phase jumps are more easily detected in the far field, because the corresponding height jump ($dz$) is determined by

$$dz = \frac{\lambda I R}{B}, \quad (3)$$

where $I$ is an integer that represents the number of misinterpreted phase cycles. From Eqs. (2) and (3), we derive that the height jump will increase at a larger factor than random noise, hence is easier to detect in the far field. In this study, the glacier front and mélange are >2 km away from the radar, and one cycle of phase jump corresponds to >58 m height jump, which is easily detected and fixed.

After applying these corrections, no time-dependent trends are found for points on rock. To assess the uncertainty of the final DEM, we chose five boxes (green, orange, olive, red, and blue boxes in Supplementary Fig. 1a) with relative steady motion but at different distances to the radar, and use the root-mean-square (RMS) deviation of elevation time series as a measure of the error. For stationary points on rock (within blue box), RMS is calculated using elevation time series subtracted from a median. For slow-moving points in the other boxes, RMS is calculated using linearly detrended elevation time series to account for melting and long-term ice motion. Supplementary Fig. 1f shows RMS for all pixels corresponding to boxes with the same colors. The light color represents the RMS of nonsmoothed time series. The dark color represents the RMS of 30-min median filtered time series. Large RMS values typically occur over areas adjacent to TRI LOS shadow (due to surface topography). Based on Eq. (2), error in elevation is proportional to both slant range $R$ and phase error $\sigma_\phi$, and phase error is associated with coherence, which generally decreases with distance (Supplementary Fig. 2a). Therefore, we surmise that uncertainty of our DEM estimates is proportional to the square of slant range distance to the radar, and use the equation

$$\text{RMS} = aR^2, \quad (4)$$

to fit the RMS vs. slant range scatter. The grey curve in Supplementary Fig. 1f is the best fitting curve to the dots with light color. Black is the best fitting curve to the scatter calculated from 30-min median filtered elevation time series. The fitting curves describe RMSs of the majority of points quite well, although they may be biased at some locations, especially at the areas adjacent to TRI LOS shadow. For the black curve in Supplementary Fig. 1f, the coefficient $a = 0.02$ km m$^{-2}$ (for convenience, $R$ has units of km in Eq. (4), while RMS has units of m). We use this for error propagation in our melt rate and ice mass loss estimates (below) since those are all based on 30-min median DEMs. Supplementary Fig. 1g shows elevation time series for representative points marked by the same color dot within corresponding boxes in Supplementary Fig. 1a, light and dark colors represent nonsmoothed and 30-min median filtered time series. Note that 30 min is a reasonable window because ice in the mélange moves at a speed of ~30–50 m d$^{-1}$, thus ice motion within a typical 30 min period is of order <0.1 pixel width (1 pixel width is 10 m in these TRI images), similar to the level of displacement uncertainty from the feature tracking method.

We also compare the uncertainty estimated above with a theoretical uncertainty model. According to the model of Rodriguez and Martin[35], the standard deviation of unwrapped phase for a single radar scan is

$$\sigma_\phi = \sqrt{\frac{1-c^2}{2c^2}}, \tag{5}$$

where $c$ is coherence, generally >0.99 for points within 8 km of the radar after adaptive filtering (Supplementary Fig. 2a)[36]. We can then estimate the uncertainty of our elevation estimates due to random noise in the phase measurements using the random term of Eq. (2). In our case, random noise for a single scan is well below 1 m for areas within ~3 km to the radar, and increases with distance (Supplementary Fig. 2b). For a 30-min average, the noise is typically below 1 m for areas within 8 km to the radar (Supplementary Fig. 2c). Supplementary Fig. 2d shows predicted uncertainty in the elevation map based on the uncertainty model derived in Supplementary Fig. 1f. Within 2–6 km of the radar, the random noise model derived from the measured elevation time series closely resembles the theoretical error from Eq. (5). At further distances, the noise model estimated from our measurements produces larger uncertainty than the theoretical model. At closer distances, the noise model derived from our measurements is smaller than the theoretical model. However, areas within 2 km of the radar are not used in calculation of ice mass loss.

Strozzi et al.[13] used a baseline of 25 cm to do topographic mapping with a TRI, appropriate for many applications, and demonstrated height precision of several meters within a distance of 2 km. Voytenko et al.[14] used a baseline of 25 cm to do repeat TRI campaign at Breiðamerkurjökull in Iceland, and found that over a stationary area (<2 km from the radar), the RMS difference between 2-h averaged DEMs from two different years is ~2 m. We use a longer baseline (60 cm), and apply additional corrections to the standard processing steps. The uncertainty of our 30-min median filtered DEMs is more than a factor of three smaller than previous studies at comparable distances and time-average windows[13,14]. In general, a longer baseline reduces random noise because baseline length is inversely proportional to phase error (Eq. (2)). However, a longer baseline can potentially induce other phase unwrapping problems because the interferograms will be more sensitive to height changes, which requires additional corrections, as described above.

**Tidally induced ice elevation change and alternate uncertainty assessment**. TRI-derived elevations in the mélange show significant tidal variations. Assuming mélange ice is floating, we can compare ice surface elevations with predicted tidal heights to give an independent assessment of precision for our elevation data.

There was no tide record in the fjord near the terminus during our TRI campaign. Previous work[5] suggests that ocean tides in the fjord within 5 km of the glacier front closely agree with tide record at Ilulissat near the mouth of the fjord, with no measurable delay in time, and the maximum difference in stage is <10 cm. We, therefore, use predicted tidal heights from the model based on long-term sea-level records at Ilulissat to represent tidal variation at the proglacial mélange, similar as Xie et al.[6]. Tidal measurements observed from a mooring at the mouth of the fjord in 2015 show no significant difference compared to tidal predictions (Supplementary Fig. 3a), RMS of the residuals is 10 cm, revealing that the tidal model works well for our purpose.

In addition to tidal-frequency signals, there are clear nontidal variations in elevation time series (Supplementary Fig. 3d, e). These nontidal variations can be caused by several factors, such as ice motion that brings icebergs with different heights into the study point, time-varying melt rate, or, deformation within the mélange, etc. To model these variations, we choose a point (D in Supplementary Fig. 3b, marked by a red X symbol) in the mélange that is ~2.8 km to the radar. No large icebergs entered this location. We use a tidal height prediction plus a second-order polynomial to model nontidal variations for each period. Periods are separated by large calving-like collapse events. The function is

$$H_{ti} = a_j + b_j t_i + c_j t_i^2 + \sum_{k=1}^{6} M_k \cos(2\pi f_k t_i + \phi_k), \tag{6}$$

where $H_{ti}$ is the observed mélange height at time $t_i$. $a_j$, $b_j$, and $c_j$ are coefficients of second-order polynomial for the $j$th period. Since the period after the last calving-

like event is too short (~9 h), we combine it with the previous period, giving 5 periods in total. $M_k$, $f_k$, and $\phi_k$ are the amplitude, frequency, and phase of tidal constituent $k$, among O1, K1, 2N2, N2, M2, and S2[37]. Note in Eq. (6) only $a_j$, $b_j$, and $c_j$ are parameters to be estimated (total number is 15). The red curve in Supplementary Fig. 3d shows the least squares best fitting curve. RMS of the residuals is 22 cm, representing a combination of uncertainties in the tidal model, nontidal variation, and TRI-derived elevations. Thus 22 cm is an upper bound of uncertainty in elevation data at this location.

Supplementary Fig. 3e shows elevation time series of a mélange point (E in Supplementary Fig. 3b marked by a red X symbol) that is close to the glacier front. In addition to tidally induced elevation changes, nontidal variations are significant. This is mainly because there are many large icebergs with varying heights near this location, thus TRI will measure a higher elevation when a higher iceberg moves into this location. While we do not attempt to model nontidal variations in this time series, to compare with tidal heights, we band-pass filter (frequencies between 0.8 and 4 cycle-per day passed) the elevation data, shown in red in Supplementary Fig. 3e. The amplitude and phase match tidal predictions well.

Another method to derive elevation change from TRI measurements was first described by Voytenko et al.[38]. This takes advantage of the TRI characteristic that displacement measurements are only sensitive in the LOS direction. Thus a point with ice flow perpendicular to the radar LOS direction should have zero displacement as seen by the radar, unless it has vertical motion (e.g., caused by tides). Point E in Supplementary Fig. 3b moves almost perpendicular to the LOS direction (within ±5 to 90°)[6] so its projection of displacement onto radar LOS should have minimal influence on observed periodic signals. Assuming the TRI-observed displacement (grey dots in Supplementary Fig. 3f) was only caused by vertical motion (Supplementary Fig. 3c describes the geometry), we can calculate vertical motion by inversely projecting the integrated LOS displacements (black dots in Supplementary Fig. 3f) onto vertical direction using

$$D_{\text{ver}} = \frac{R}{H_r} D_{\text{los}}, \tag{7}$$

where $D_{\text{ver}}$ is integrated vertical displacement (tidal variation). $R$ is slant range distance from the radar to the study point, ~2.8 km. $H_r$ is the height difference between the radar and the study point, ~190 m. $D_{\text{los}}$ is integrated LOS displacement measure by TRI. These parameters are shown in Supplementary Fig. 3c.

Red dots in Supplementary Fig. 3f show elevation time series (frequencies between 0.8 and 4 cycle-per day passed) from Eq. (7). The overall amplitude of these tidal estimates is significantly larger than the tidal model, and the phase difference is also larger than the extracted tides from elevation data shown in Supplementary Fig. 3e. We interpret these as side-to-side motion of ice in the mélange, which might be related to ocean currents[6]. Hence, the assumption that the TRI-observed LOS displacement at this location is only caused by vertical motion is invalid. However, this method may still be useful to provide tidal information in the absence of other data[38]. In addition, this method requires only one receiving antenna.

**Mean mélange thinning due to divergence and melting**. Due to spatially non-uniform motion and fjord dimensions, mélange ice diverges, leading to surface elevation change independent of melting. Figure 6 and Supplementary Fig. 6 show example velocity and displacement fields estimated by feature tracking. Mélange ice near the south bank generally moves slower than ice in the middle of the main trough, likely affected by curvature of fjord wall. We calculate divergent ice motion by treating each pixel within the selected box as an independent cell, and assume ice displacement is constant with depth. Area changes, which are related with ice divergence, are calculated using the convex hulls determined by all cells using the Python Qhull library[39]. This allows us to determine the mean divergence within the selected box. Note that we treat mélange as incompressible material, and we do not try to solve for divergence of each pixel independently, because a meaningful divergence can only be estimated in a Lagrangian reference frame given the fast moving mélange. Thus all ice elevations derived after the reference time (which defines initial positions and heights for all cells) are linearly correlated with divergence. Mean divergent thinning rate ($H_{\text{mdr}}$) can be calculated with

$$H_{\text{mdr}} = \left(\frac{A_1}{A_0} - 1\right)\frac{H_{m1}}{\Delta t}\frac{\rho_w}{\rho_w - \rho_i}, \tag{8}$$

where $A_0$ and $A_1$ represent the area of the box before and after divergence separated by time $\Delta t$. $H_{m1}$ denotes the mean ice surface elevation mapped by TRI (tide detrended). $\rho_w$ is water density (1027 kg m$^{-3}$); and $\rho_i$ is ice density (917 kg m$^{-3}$). The mean divergent thinning rate in the selected Lagrangian box (Fig. 5) is used to calculate the mean melt rate ($H_{\text{mr}}$)

$$H_{\text{mr}} = \frac{H_0 - H_1}{\Delta t}\frac{\rho_w}{\rho_w - \rho_i} - H_{\text{mdr}}, \tag{9}$$

where $H_0$ and $H_1$ denote mapped ice surface elevation separated by $\Delta t$. In Eqs. (8) and (9), uncertainties in $A_1$ used for error propagation are simplified to the uncertainties in the two orthogonal axes of the corresponding cover hulls determined by feature tracking. We assign no error in $A_0$, hence the resulting uncertainty could be underestimated. Uncertainties of $H_{m1}$, $H_0$, and $H_1$ are calculated with the uncertainty model defined in Supplementary Fig. 1f, however, local sea

level defined by us (Supplementary Fig. 1c, d) may cause an offset of these values. Ice density can also differ due to variable compaction, while surface water density can differ due to changes in salinity, mélange may not be perfectly incompressible. However, these will not change the signs of Eqs. (8) and (9) or adversely change the mechanisms of overall ice loss in the mélange, and will not affect our major conclusions.

**Glacier calving rate**. Glacier calving events are determined by inspection of TRI, Landsat-8, and Sentinel-1/2 images. Figure 3 lists selected images to show ice loss due to glacier calving between 5 May and 29 June 2016. Before and after the period shown in Fig. 3c–l, there are additional calving events visible on satellite images. We chose this period to derive a relation between changes in the mélange and glacier calving because we have TRI observations within this period.

To estimate ice loss due to calving, we first digitize calving front positions by manually drawing locations of the glacial front on different images. 3-pixel width (45 m) in a Landsat-8 panchromatic image is used to estimate uncertainty in the position estimates. This includes errors caused by digitizing and geolocation. From 21 May 2016 until the end of TRI observations, only one minor glacier calving event was detected (Fig. 3f, g). Large calved areas are found after the TRI observations (Fig. 3j–l).

The area of glacier ice loss has two components: first, changes between the digitized ice cliff positions, shaded with blue or red in Fig. 3f, k; second, ice that was out of the shaded areas before the calving events but later falls within the shaded areas due to ice motion (ideally, calved ice area should be estimated in a Lagranian reference frame). The first part accounts for the majority of total glacier ice loss during our study period. The second part accounts for ~10% because ice near the glacier front moves fast (>30 m d$^{-1}$). Ice loss due to this component is calculated using velocities from feature tracking and is shown in Supplementary Fig. 6.

To convert area of ice loss into mass of ice loss, we assume an average ice thickness of 800 m. This is based on measured surface elevation (~100 m) (Fig. 2) and the depth of bed bathymetry (~600–1200 m)[18] near the glacier front. For the period 20–29 June, total calved ice mass is 6.7 ± 0.8 Gt. For reference, average annual ice mass loss for all of Greenland for the decade 2003–2013 was 244 ± 6 Gt ($2\sigma$)[30]. A different ice thickness will change the values of calved ice mass and calving rate (we define it as calved ice mass per day), but would not change the inverse relation between mélange ice mass and glacier calving rate (Fig. 7e).

The mass of the wedge of mélange ice in front of the glacier grows by calving, and shrinks by downstream advection and divergence of ice, gravitational collapse of elevated ice at the toe of the wedge, and sub-aerial and submarine melting. Due to possible feedback between mélange mass or strength and glacier calving, the study period may represent one of several mélange-glacier mass variation cycles in the late spring and summer melt season.

**Speed and strain rate changes**. Supplementary Fig. 7a, c, e shows examples of speed changes before and after major calving events, estimated by feature tracking. Supplementary Fig. 7b, d, f shows longitudinal strain rates during different periods calculated by using the logarithmic strain-rate calculation code of Alley et al.[40], with an effective length scale of 300 m. Different length scales do not change the overall pattern but longer length scales yield smoother strain rate maps. Despite changes in mélange thickness, terminus position, and possibly floatation status, speed and longitudinal strain rate in the middle of the glacier show no significant increase. By the end of the TRI observation period, the glacier front near the coast (to the radar side) has higher speed and longitudinal strain rate compared to the beginning of the TRI observation period, corresponding to the area with newly formed fissures by the end of the TRI observations (Fig. 9d and Supplementary Fig. 8), suggesting a decrease in buttressing force from mélange downstream from the elevation step-change.

The longitudinal and transverse strain rates also provide a way to estimate divergence thinning rate. For the dashed cyan area in Supplementary Fig. 7b, d (also shown in Fig. 5), we estimated a divergence thinning rate of 0.04 m/d on the first TRI observation day, and a divergence thinning rate of 1.84 m/d on the last TRI observation day. These are comparable to divergence rate estimates shown in Fig. 5n, and indicate an overall increase in ice mobility of the mélange.

Supplementary Figure 8 shows LOS strain rate changes throughout the TRI observation period. For two points (P1 and P2) separated by a distance of $d$ along one radar LOS, in a time interval of $\Delta t$, unwrapped phases at P1 and P2 changed by $\Delta\phi_1$ and $\Delta\phi_2$, respectively, then LOS strain rate between P1 and P2 during $\Delta t$ is calculated as

$$\dot{\epsilon}_{\mathrm{los}} = \frac{\lambda(\Delta\phi_2 - \Delta\phi_1)}{2\pi d \Delta t}, \qquad (10)$$

where $\lambda$ is the radar microwave length (1.74 cm). Note the constant high and low zones (dark blue) can be caused by high gradients in LOS velocities due to geometry effects. However, changes in LOS strain rate should represent real changes in strain rates. During the observation period, newly formed high LOS strain rate zones (marked by cyan arrows) occurred immediately after calving-like collapses, located upstream from the elevation step-changes.

**Mélange buttressing force estimate**. Two approaches have been used to estimate mélange buttressing force decrease, described below.

In the first method, if we consider the mélange wedge as a weak ice shelf[10,20] that is an extension of the glacier, then buttressing from the downstream thin mélange is a resistive force that prevents or limits calving-like collapses within the upstream mélange wedge, and also acts on the glacier front. The change of buttressing force at the elevation step-change is thus a lower bound estimate of change at the glacier front. We assume that the buttressing force from the downstream mélange will decrease to a low value (and possibly zero) immediately after each collapse and then increases until the next collapse. This is supported by our data: ice speed immediately upstream from the elevation step-change increases stepwise after calving-like collapses (Fig. 8e1); new extensional fissures formed in the mélange wedge immediately after each calving-like collapse (Fig. 9); DEMs here show that ice thickness immediately downstream from the elevation step-change is a minimum (Fig. 10a and Supplementary Fig. 4f). With this assumption, the buttressing force from mélange downstream from the elevation step-change during periods of mélange quiescence approximates the buttressing force reduction acting on the remaining thick mélange wedge immediately after each calving-like collapse. This also represents buttressing force reduction on the glacier front immediately before major calving on 20 or 21 June 2016, assuming the thick mélange wedge had retreated to a minimum by then. We do not have TRI or satellite data immediately before major calving events during this 1.5-day period, however, by the end of our TRI observations, the elevation step-change was ~2 km from the glacier front and extensional fissures formed immediately in front of the glacier, suggesting that further calving-like collapses in the mélange would soon occur. In support of this, Amundson et al.[10] found that larger calving tends to occur after the mélange was pushed away from the terminus of Jakobshavn Isbræ by small calved icebergs.

Using the model of Burton et al.[20], we can calculate the buttressing force per unit lateral-width on the glacier front from the relation

$$F_{\mathrm{m}} = \frac{\sigma_0 H_{\mathrm{m}}}{\mu}\left(e^{2\mu L/W} - 1\right), \qquad (11)$$

where $F_{\mathrm{m}}$ is the buttressing force per unit lateral-width from the mélange acting on the glacier front, $\sigma_0$ is the minimum shear stress required to produce flow through rearrangement of mélange particles (8.25 kPa[20]), and $\mu$ is the effective coefficient of friction, depending on the material friction coefficient and the geometry of the fjord walls (0.3[20]). $L$, $W$, and $H_{\mathrm{m}}$ are the length, width, and thickness of the mélange (Supplementary Fig. 9a). Based on satellite images of the fjord and our data, we simplify the mélange as a cuboid 7.7 km wide, 31.0 km long, and 39.6 m thick (Supplementary Fig. 9). The thickness is an average of all pixels within the dashed polygon M in Supplementary Fig. 9b, downstream from the elevation step-change. We choose this polygon to approximate the average thickness of the simplified mélange cuboid because this area is covered by TRI measurements, and is downstream from the elevation step-change of the mélange, hence better represents a broad area of mélange. The height datum of TRI-derived DEMs is defined by 2% of measured heights and can have an offset to the true local sea level (described above), thus our ice thicknesses may be underestimated. The 39.6 m mélange thickness estimate used here is significantly smaller than mélange thickness measured at other fjords[32–34]. Using the above parameters in Eq. (11) yields a buttressing force per unit lateral-width of $1.1 \times 10^7$ N m$^{-1}$ exerted from downstream thin mélange prior to calving-like collapse events. Here, we do not count the contribution of the thick mélange wedge, as it has only a minor effect on the length-to-width ratio used in Eq. (11). However, while the effect of wedge structure is not counted in this model, it presumably increases the buttressing force at the glacier front. Thus the above value is likely a minimum estimate of buttressing force decrease at the glacier front immediately before major calving began on 20 or 21 June 2016.

In the second method, assuming that the mélange is homogeneous over its thickness and total buttressing force applied on the calving face is proportional to mélange thickness[21], we can estimate the decrease in buttressing force if buttressing force exerted per unit thickness of the mélange is known. Since mélange thickness is not uniform here, we use an average thickness over an area immediately in front of the glacier (polygon P in Fig. 7a) to represent the effective buttressing thickness of mélange. At Store Gletscher ~140 km north of Jakobshavn Isbræ, the buttressing stress on the entire glacier calving face ($\sigma_g$) due to mélange was estimated at ~30–60 kPa[31]. To our knowledge, this is the only direct estimate of mélange buttressing stress at the glacier front. This range of values has been implemented in calving models for both Store Gletscher in Greenland[32] and Hansbreen Glacier in Svalbard[41], despite their different settings. In the following calculation, we also assume that it is representative of Jakobshavn Isbræ.

The estimate of ~30–60 kPa buttressing stress corresponds to the entire thickness of the glacier front in the longitudinal coupling model of Walter et al.[31]. The actual buttressing comes from a smaller mélange-glacier contact surface. At Store Gletscher the buttressing stress on the entire thickness of the glacier front is equivalent to ~240–480 kPa mélange-glacier contact pressure[32] using

$$\sigma_{\mathrm{m}} = \sigma_{\mathrm{g}} \frac{H_{\mathrm{g}}}{H_{\mathrm{m}}}, \qquad (12)$$

where $\sigma_{\mathrm{m}}$ is the buttressing stress on the mélange-glacier contact. $H_{\mathrm{g}}$ and $H_{\mathrm{m}}$ represent the thickness of glacial front and mélange, respectively. For Store Gletscher, $H_{\mathrm{g}} = 600$ m[31] and $H_{\mathrm{m}} = 75$ m[32]. Using average mélange thicknesses in our test area (Polygon P shown in Fig. 7a), the buttressing force per unit lateral-

width applied on the calving face by the mélange is[21]

$$F_m = \sigma_m H_m. \tag{13}$$

During the ~13 day TRI observation period, average mélange thickness of the test area decreased by ~37 m. Substituting $H_m$ in Eq. (13) with the thickness decrease yields a total force decrease applied on the glacier front of ~$0.9$–$1.8 \times 10^7$ N m$^{-1}$ (force per meter of lateral-width), equal to a ~11–22 kPa pressure change upon the entire glacier front assuming it has a thickness of 800 m. Further calving-like collapse events within the mélange between 12:20 20 June 2016 (last TRI observation) and 00:00 22 June 2016 (first Landsat-8 image acquisition following the end of the TRI observations) may have reduced the buttressing force even more. If the entire mélange wedge was moved away by further calving-like collapses before major calving events, the total decrease of mélange thickness in the test area is ~89 m, yielding a total decrease of buttressing force by ~$2.1$–$4.3 \times 10^7$ N m$^{-1}$ before major calving events after the TRI observation period, equal to a ~27–54 kPa pressure change upon the entire glacier front assuming it has a thickness of 800 m. Since glacier speed did not change significantly during the TRI observation period, it is possible that most of the buttressing force decrease acting directly at the glacier front occurred after removal of the remaining mélange wedge (within 1.5 days after our TRI observations). Perhaps buttressing force acting on the glacier remained largely unchanged during the TRI observations, reflecting rapid decay of stress transmission with distance, but buttressing force acting on the newly formed elevation step-change dropped significantly and shear stress at the constricting margin of the remaining mélange wedge increased. Once it reached a threshold (yield stress), the remaining mélange wedge failed, the glacier front became the new elevation step-change, buttressing force on the glacier calving face dropped to very low values, and large calving events occurred. Note that the average mélange thickness of our test area (Polygon P shown in Fig. 7a) may not be the best estimate of effective buttressing thickness, or properly account for the effects of wedge geometry. Thus the calculated ~$0.9$–$1.8 \times 10^7$ N m$^{-1}$ is likely a minimum estimate of the decrease in buttressing force from the beginning of TRI observations to the first major calving event. Further research is needed to quantify these effects.

## Data availability

Landsat images were downloaded through the USGS EarthExplorer (https://earthexplorer.usgs.gov/). Sentinel-1/2 data provided by European Space Agency and were downloaded through the USGS EarthExplorer and Alaska Satellite Facility (https://www.asf.alaska.edu/). TRI data (>2TB) are available upon reasonable request. Additional data can be found in the supplementary figures and movies. Programs and methods presented in this paper are either noted using references to their sources or provided with detailed equations.

## Code availability

The codes used for data analysis and figure plotting are available from the corresponding author upon reasonable request.

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

## Acknowledgements

Denise Holland at the Center for Global Sea Level Change in New York University Abu Dhabi organized the field logistics. Fanghui Deng at the University of South Florida is thanked for helpful discussions. This research was partially supported by NASA grant NNX12AK29G to T.H.D. D.M.H. acknowledges support from NYU Abu Dhabi grant G1204, NSF award ARC-1304137, and NASA Oceans Melting Greenland NNX15AD55G. ArcticDEM provided by the Polar Geospatial Center under NSF-OPP awards 1043681, 1559691, and 1542736.

## Author contributions

S.X. analyzed the data and wrote the paper together with T.H.D. D.M.H. organized the TRI campaign and started a discussion that led to this research. D.V. operated the radar and helped in the initial data analysis. I.V. participated in discussions. All authors commented on the paper.

## Additional information

**Competing interests:** The authors declare no competing interests.

