## [Peer Review File · Nature Communications]

Reviewers' comments:

Reviewer #1 (Remarks to the Author):

Review of "Rapid iceberg calving following removal of tightly packed pro-glacial mélange"

This study details observations of iceberg mélange and associated iceberg calving dynamics at Jakobshavn Isbrae in Greenland. A terrestrial radar interferometer (TRI) is used to measure mélange and glacier heights and motion in high temporal and spatial detail. Over a short observation period (13 days) in summer 2016, a thick and tightly packed "wedge" of mélange was observed near the front of the glacier that appears to have suppressed calving. Once this wedge of thick mélange reduced in mass, iceberg calving began again.

Over a 10-year period at this glacier, no other window of time showed a similar 30-day shutdown of calving. This is an important observation, and indicates that a buildup of a thick mélange wedge can suppress calving. However, this finding is not really clear from the main text. Only after a careful read of the Supplementary Information is this clear. The SI is very long and technical, and could itself be an independent methods-based paper. Unfortunately some of the context and implications of this study are buried in the SI, leading to some confusion in the main text.

Figure S7 is one of the most important figures for placing the observations and conclusions of this study in context. It indicates that the observed period of suppressed calving (~30 days) is unique over a 10-year period. However, this also suggests that the observed thick mélange wedge might be an anomaly, and not very important to the long-term evolution of the glacier and associated calving losses. After the thick wedge retreated closer to the glacier front, calving began to proceed again, but not at an apparent rate that was unique compared to other years at the same time. Thus it seems that the authors observed a peculiarity. These implications aren't really discussed in the text, but it seems essential to address. Did the authors just happen to catch a rare, albeit important, time period of thick and buttressing mélange? This seems to be the case. The findings may suggest that you need very thick mélange (more thick than usual) to suppress calving at this particular glacier. The results are still novel and interesting, but I'm not sure how important they are for justifying publication in a high-impact journal.

The figures in the paper are rather confusing to interpret. There are many different choices of colormaps, all of which are rather difficult to interpret, and the captions are difficult to follow as since the different sub-panels are referenced in multiple places in most of the captions.

Some line-by-line comments:

L 5: “mélange” is usually defined in the glaciological literature as a mixture of icebergs, sea ice, and snow. The term “compressed ice fragments” is inconsistent with this definition, and somewhat misleading. The term “compressed” has a different connotation in mechanics, and “fragments” is not a term that would usually be associated with large icebergs.

L 8: quantitative understanding of what?

L 10: improvement in precision of what?

L 14: it’s unclear at this stage what you mean by “wedge” here. This could be interpreted as a plan-view wedge or a cross-sectional wedge (it becomes clear later, but at this stage the description is vague)

L 21: “likely because that” is awkward

L 36: “terminus is now embedded within the ice sheet” is strange phrasing, and a bit confusing

L 42: “sufficiently large”

L 49-50: it would be helpful to be a bit more specific about what this approach entails. Without diving into the details in the main text, you can still describe generally what you did to achieve a higher precision.

L 55: the term “precision” is used in many places in the text, but it’s not clear if you don’t sometimes mean “accuracy”. In this case, you’re describing the comparison of your TRI-derived DEM with other stationary points and tide predictions, which sounds like more of a check of accuracy.

L 55-56: “median average” is confusing, as median and average are different things. Are you talking about a median of averages? And of what? The kernel size is meaningless without context of what you are talking about.

L 59-60: What are these new processes? It would be helpful to state them here.

L 75: A tightly-packed granular material does not necessarily mean high cohesive strength, nor does it need to have cohesive strength at all to provide buttressing resistance to motion.

L 76: I don't think you can substantiate the “unusually tightly packed” claim here. Your 13-day observation window cannot support this claim. The unusual observation is that there was no observed calving when you observed the thick wedge of melange, but you have no other observations of melange thickness or packing to compare against.

L 79: 2016? or are you still referring to 2015 here?

L 193: “mort”?

L 228-230: you should really describe what these properties and assumptions are. Don't expect the reader to dive into all these external references to figure out what you did.

Figure 2: it would be helpful to draw the grounding line in for reference. The colormap is rather strange and distracting, as it draws the eye to the higher elevations with strange and strongly varying colors far from the area of interest. Perhaps mask out the grounded ice and use a perceptually uniform colormap.

L 325-326: but the red lines are over the blue lines, not the black dots... something is wrong in your figure or description here.

L 327: so the blue and red curves are shifted?

Figure 3: different colormaps here from that in the previous figure (why?), and the "jet" or "rainbow" colormap is not appropriate for sequential data, as it is a divergent colormap (and not perceptually appropriate anyway; very bad for colorblind people, and even for non-colorblind people it leads to incorrect interpretations of the data where the color luminance strongly varies).

Figure 4: another change of colormaps. Red and green together in these maps is very hard to discern for colorblind people. It also gives the false impression of a step-change at the color change, even if the variable you are representing changes gradually. A perceptually uniform colormap would be much more appropriate to use.

Figure 4e: pretty short timescale of observations... how does this fit into the longer term perspective? did you observe something usual or unusual? this is important to address to judge the implications of your findings (this is eventually made clear in the Supplementary Material, but only for the VERY astute reader, as it is buried rather deep...)

L 346: "bottom heights" refers to inferred bottom heights presumably? versus surface heights, which is what you measured.

Supplementary Information: it would be helpful to have figures in line with the text, it is very tedious to scroll between the figures and the text describing them here. In the absence of line numbers, it is tedious to include comments on specific line numbers for further minor comments. Similar comments for figures (colormaps and captions) as for the main text.

Reviewer #2 (Remarks to the Author):

This manuscript described novel observations of the ice mélange in front of the calving terminus of Jakobshavn Isbrae, Greenland. The authors use a new radar-based approach to map time-varying elevations near the mélange-glacier interface. The main conclusion is reiterated in the title: removal of tightly packed pro-glacial mélange results in rapid iceberg calving.

This conclusion is not supported by the measurements presented, and thus my recommendation is that the manuscript be rejected.

There are two main problems with the paper. First, while step-change in mélange thickness shown in Figure 2b migrates somewhat towards the glacier terminus, the mélange thickness immediately in front of the terminus actually increases quite substantially. Figure 2e shows an increase of about 15 m at point E. Assuming the mélange is floating, this corresponds to a thickening of more than 100 m. Thus, if the back stress exerted by the mélange is indeed proportional to the thickness, this thickening would lead to a substantial increase in back stress and thus further reduction in iceberg calving. I fail to see how the much thinner mélange beyond the step change in elevation can overcome this back pressure.

The second problem is more serious and seems to point at some problem with the data or data processing. The speed of Jakobshavn is around 50 m/day so over the three-week observation period, the terminus and mélange should have moved forward by 1 km or so. Yet the elevation maps in Figure 3 show no movement of the mélange. The three color maps shown in panels d-f are nearly identical, with only minor elevation changes. Similarly, the first movie provided in the SI shows no movement whatsoever of the glacier terminus region or of the thicker mélange in front of the terminus (except for three jerky adjustments of the whole region).

I find it hard to believe that the calving rate of the glacier is zero throughout the period and jumps up on June 16 – when the observations have been terminated. If indeed the calving rate were zero from May 20 to June 19, the terminus should have advanced by 1.5 km. This advance is not shown in the radar observations – or I must be missing something.

Reviewer #3 (Remarks to the Author):

This study uses a terrestrial radar interferometer to quantify changes in the mélange thickness extending in front of Jakobshavn Isbrae, Greenland. This mélange has been hypothesized to exert a buttressing force that can stabilize and inhibit calving from Jakobshavn and other marine terminating glaciers. Understanding the evolution of the pro-glacial mélange along with its mechanical interaction with the upstream glacier has thus been hypothesized to be an important piece of the ice dynamics puzzle. In this study, the authors are able to resolve the evolution of the height of mélange (or thickness, assuming freeboard) and to examine how this influences calving. Overall, this is a novel study that provides one of the first high resolution (temporal and space) examinations of a relatively newly discovered process. This type of data is likely to be of increasing interest to the glaciological community.

I did, however, have a few questions and comments. First off, I think one of the strong points of the paper is the attempt to examine different processes that control the evolution of melange. To my understanding (and I apologize if I misunderstood), the authors try to parse out how much of the melange thickness change is due to melting and how much is due to the divergence of melange. The calculation is briefly described in the text and supplementary material, but it appears to assume that the melange can be treated as an incompressible fluid. This seems like a questionable assumption given the blocky and granular nature of melange described by the authors. Here, I think it would be helpful to more clearly state the assumptions of the calculations performed along with an honest assessment of any limitations.

Second, and perhaps related, the authors seem to assume that melange can be treated as a fluid and thus it can form an ice shelf. If this is the case, why not try to fit an analytic ice shelf profile (e.g., Van der Veen *Fundamentals of Glacier Dynamics* p. 132, Equation 5.71) to see if the ice shelf really behaves like an ice shelf. The advantage of this approach is that the authors can infer the effective stiffness of ice associated with melange. A distinct possibility is that the melange does not behave like an ice shelf and instead the wedge is more like a compressional wedge. A good example of this is the wedge of sand that is formed in front of a bulldozer. In this case, the ice shelf is the bulldozer and it is ploughing through the melange causing it to compress in front. The compression is the result of the motion of the calving front pushing melange against pinning points and the thickness of the wedge in this case is the result of compression (often formed through jamming), but not the source of the compressive resistive stresses.

Third, it is interesting that the thickness of the melange is not correlated at all with the glacier velocity (or, presumably, the strain rate). Given the fact that the authors are attempting to estimate buttressing, it is straightforward to use that buttressing to estimate the effect it would have on the strain rate near the calving front (the authors cite several examples where this is done). This would give a sanity check on the magnitudes of buttressing estimated and ensure they are consistent with ice dynamics. Moreover, it is possible that the buttressing is not inhibiting the dynamics of the glacier (consistent with the lack of an effect on the velocity), but instead is acting to suppress bergs that have already calved from detaching. Both are possible and teasing out which is actually happening would be an important advance.

There is also some loose terminology in the paper that I found hard to understand. For example, what is a calving-like collapse and how is it different from a calving event? I think a little bit of explanatory text would help readers understand what the authors mean. I had similar questions

about the step-step change “migrating” closer towards the calving front. To me, it seems as though the melange is breaking up and the evacuation of melange is causing the wedge to change shape.

Finally, I had significant problems interpreting the figures. Here, I would encourage the authors to adopt color schemes that are useful. There are numerous guides to assist this. For example, this webpage <https://knightlab.northwestern.edu/2016/07/18/three-tools-to-help-you-make-colorblind-friendly-graphics/> is quite helpful. Perceptually uniform color schemes are also a must. Figure 2, I suspect is an example where a better color scheme would make the figure informative (I see multiple low velocities, which I suspect are purely artifacts of the badly chosen color scheme). I highly recommend the authors use a tool (e.g., <http://www.colororacle.org>) to examine how their color schemes are likely to be perceived by people with different eye conditions (color blindness is very common). I honestly see very little in the colored portions of Figures 2-4 and have largely ignored these figures. More color perceptive people may have a different reaction to these figures.

Detailed comments:

Abstract: Typically abstracts for Nature publications are fully referenced.

Page 2, line 19-20: I thought that surface melt now accounts for over half of the mass lost from Greenland?

Page 3 lines 43-44. It is misleading to argue that the effect of pro-glacial buttressing needs to be accounted for in models. Whether buttressing is important depends on the time and spatial scales considered—melange is probably not of great importance for ice sheet of models of glacial-interglacial cycles. Moreover, most glacier models fail to accurately resolve terminus motion in the absence of melange. In these cases, it is hard to see how “accurately quantifying” melange buttressing would significantly improve model capability. A more limited, but accurate statement might be that understanding the observed evolution of Greenland outlet glaciers requires a better understanding of the role of melange.

Page 3, line 53: Shouldn't error be plural? “We corrected height error*s*”

Page 3, line 56: Shouldn't minutes be singular? “30 minute** window.

Page 4, line 69: Are these calving events? Or do these take place in the melange?

Page 4, line 75: Why does a lack of calving events indicate the melange is tightly packed? There are *many* glaciers that calve less frequently than 17 days with little to no melange. Why is this not a property of the glacier? The other lines of evidence are more convincing.

Page 5, lines 104-105. The estimate of melting appears to assume that melange is incompressible. This seems like a highly questionable assumption that needs to be either justified or, better yet, quantify the error associated with this assumption. See my previous comment.

Page 8, line 166, word "thickness" is repeated.

Page 8, line 176. I'm assuming the "s" after 10 is means to indicate tens. I think this would be more clear if the authors just used a squiggly line to indicate approximately equal to.

Figures.

Figure 1. The orientation of panels a and b was hard for me to parse. It looks to be like Panel a is rotated relative to panel b, but this is hard to tell. It would be nice to see both figures oriented in the same direction (or with a North arrow in both to show how the images have been rotated).

Figure 2a. It would be helpful to mark the calving front in Figure 2 a.

Figure 3: I don't understand what this figure is supposed to show. What is causing the strong variations in color?

We appreciate comments from the reviewers on our manuscript. Comments are below in black, and responses we made are in red.

Responses to Reviewer #1:

This study details observations of iceberg mélange and associated iceberg calving dynamics at Jakobshavn Isbrae in Greenland. A terrestrial radar interferometer (TRI) is used to measure mélange and glacier heights and motion in high temporal and spatial detail. Over a short observation period (13 days) in summer 2016, a thick and tightly packed “wedge” of mélange was observed near the front of the glacier that appears to have suppressed calving. Once this wedge of thick mélange reduced in mass, iceberg calving began again.

Over a 10-year period at this glacier, no other window of time showed a similar 30-day shutdown of calving. This is an important observation, and indicates that a buildup of a thick mélange wedge can suppress calving. However, this finding is not really clear from the main text. Only after a careful read of the Supplementary Information is this clear. The SI is very long and technical, and could itself be an independent methods-based paper. Unfortunately some of the context and implications of this study are buried in the SI, leading to some confusion in the main text.

We have moved important information from the SI to the main manuscript. There are 10 figures in the revised manuscript, including 10-year calving events inferred from TRI and satellite images.

Figure S7 is one of the most important figures for placing the observations and conclusions of this study in context. It indicates that the observed period of suppressed calving (~30 days) is unique over a 10-year period. However, this also suggests that the observed thick mélange wedge might be an anomaly, and not very important to the long-term evolution of the glacier and associated calving losses. After the thick wedge retreated closer to the glacier front, calving began to proceed again, but not at an apparent rate that was unique compared to other years at the same time. Thus it seems that the authors observed a peculiarity. These implications aren't really discussed in the text, but it seems essential to address. Did the authors just happen to catch a rare, albeit important, time period of thick and buttressing mélange? This seems to be the case. The findings may suggest that you need very thick mélange (more thick than usual) to suppress calving at this particular glacier. The results are still novel and interesting, but I'm not sure how important they are for justifying publication in a high-impact journal.

We have moved this figure to the main manuscript. It is Figure 4 in the revised manuscript.

We agree with the reviewer that the observed thick mélange wedge could be an anomaly. However, this unusually thick mélange and the long period of glacier quiescence without major calving events provides a unique opportunity to test the hypothesis that mélange buttressing the glacier front can suppress calving. Besides, we found no previous studies that have quantified this effect, or described the “wedge” structure we describe in our paper. With this study, we suggest the possibility that such wedges could be widespread in tidewater glaciers, but missed in

previous studies because of data limitations. Hopefully, our paper will stimulate other scientists to be aware of these structures and look for them in future studies of tidewater glaciers.

The figures in the paper are rather confusing to interpret. There are many different choices of colormaps, all of which are rather difficult to interpret, and the captions are difficult to follow as since the different sub-panels are referenced in multiple places in most of the captions.

We have changed the color maps, now all color maps are in blue-yellow-red scheme. We have also modified the captions to improve clarity.

Some line-by-line comments:

L 5: “mélange” is usually defined in the glaciological literature as a mixture of icebergs, sea ice, and snow. The term “compressed ice fragments” is inconsistent with this definition, and somewhat misleading. The term “compressed” has a different connotation in mechanics, and “fragments” is not a term that would usually be associated with large icebergs.

We have changed the context in the parentheses to “a mixture of sea ice, icebergs, and snow”

L 8: quantitative understanding of what?

We have added “buttressing effect”.

L 10: improvement in precision of what?

“elevation precision”. Added.

L 14: it’s unclear at this stage what you mean by “wedge” here. This could be interpreted as a plan-view wedge or a cross-sectional wedge (it becomes clear later, but at this stage the description is vague)

We have changed it to “mélange wedge that increased in thickness towards the glacier front”.

L 21: “likely because that” is awkward

We have changed it to “because of the ice-ocean interface”.

L 36: “terminus is now embedded within the ice sheet” is strange phrasing, and a bit confusing

We have changed “embedded within” to “embedded in”. And we have referred to “(Figure 1b)” to reduce confusion — Figure 1b shows clearly that the glacier terminus is embedded in the Greenland ice sheet rather than being at the end of a long glacier tongue.

L 42: “sufficiently large”

Done.

L 49-50: it would be helpful to be a bit more specific about what this approach entails. Without diving into the details in the main text, you can still describe generally what you did to achieve a higher precision.

We have added more detail about the processing methods of DEM within the main text.

L 55: the term “precision” is used in many places in the text, but it’s not clear if you don’t sometimes mean “accuracy”. In this case, you’re describing the comparison of your TRI-derived DEM with other stationary points and tide predictions, which sounds like more of a check of accuracy.

We are aware that we have actually assessed both accuracy (statistical bias) and precision (statistical variability), and we have made changes accordingly.

L 55-56: “median average” is confusing, as median and average are different things. Are you talking about a median of averages? And of what? The kernel size is meaningless without context of what you are talking about.

We have changed this sentence to “based on a median filtering (30-minute time window)”.

L 59-60: What are these new processes? It would be helpful to state them here.

We have added “allowing us to see new processes within the critical region of the pro-glacial mélange, such as mélange melting, collapses and tidal-induced elevation changes.”

L 75: A tightly-packed granular material does not necessarily mean high cohesive strength, nor does it need to have cohesive strength at all to provide buttressing resistance to motion.

We have removed “with high cohesive strength”.

L 76: I don’t think you can substantiate the “unusually tightly packed” claim here. Your 13-day observation window cannot support this claim. The unusual observation is that there was no observed calving when you observed the thick wedge of mélange, but you have no other observations of mélange thickness or packing to compare against.

Multiple evidences support the mélange wedge was unusually tightly packed: 1) there was an unusually long period without major calving; 2) small collapse events in the mélange caused significant downstream mélange motion but not in the mélange wedge; 3) during 2015 observation period here, smaller calving events changes the entire mélange in front of the glacier, while in 2016, a small calving event did not change nearby mélange. If the mélange was not tightly packed, lateral fraction will be small and the gravitational potential energy of the small calving ice block would greatly accelerate surrounding mélange. Note that Amundson et al. (2010) recorded calving events during an >1 year period at Jakobshavn Isbræ and they found that “at the onset of a calving event the entire lateral width of the mélange rapidly accelerates away from the terminus, even if the event onset only involves a small portion of the terminus”. So our

observation of a steady mélange wedge is a strong evidence of an unusually tightly packed mélange.

We compared our observations to previous studies and other years TRI observations, no previous studies have documented a >400 m thick mélange in front of the glacier.

We have made changes to clarify this.

L 79: 2016? or are you still referring to 2015 here?

We are referring to 2015 here. This is another line of evidence to support a tightly packed mélange. At the same season (early to middle June) in 2015, we conducted another TRI campaign at Jakobshavn Isbræ and very small ice blocks falling from the glacier caused entire pro-glacial mélange motion.

L 193: “mort”?

Changed to “more”.

L 228-230: you should really describe what these properties and assumptions are. Don't expect the reader to dive into all these external references to figure out what you did.

We have moved the method and assumptions for back-force decrease estimate to the main manuscript.

Figure 2: it would be helpful to draw the grounding line in for reference. The colormap is rather strange and distracting, as it draws the eye to the higher elevations with strange and strongly varying colors far from the area of interest. Perhaps mask out the grounded ice and use a perceptually uniform colormap.

We have added a dashed red line to show the glacier terminus. Due to limited data quality, only a small portion of the grounding line can be determined (Xie et al., 2018) and we don't show it here. Instead, the glacier front location is shown by dashed red line. We would like to keep the grounded portion since that is part of our data and may be interesting to some people.

We have changed the colormap to a nearly perceptually uniform blue-yellow-red scheme. However, we keep the “gist_ncar” (navy-green-blue-cyan-lime-yellow-red-purple-white) colormap in the supplementary Movie S2 because we think it is more informative to describe the elevation step-change in the mélange. We have compared many colormaps (>30) and feel the blue-yellow-red scheme is the best if we must avoid using green and red on the same map. It also allows a wider range of changes to be distinguished, which makes the map more informative, especially for the elevation step-change in the mélange.

Here is a comparison of the colormap we use (a) and two other perceptually uniform colormaps (b,c) for the elevation map (Fig. 2a in our manuscript), we prefer (a):

L 325-326: but the red lines are over the blue lines, not the black dots... something is wrong in your figure or description here.

Red lines are band-pass filtered data, the long-term trends have been filtered out, so they are not over the black dots. We have changed the caption to clarify this.

L 327: so the blue and red curves are shifted?

Yes they have been shifted for clarity. We have changed the description to clarify this.

Figure 3: different colormaps here from that in the previous figure (why?), and the "jet" or "rainbow" colormap is not appropriate for sequential data, as it is a divergent colormap (and not perceptually appropriate anyway; very bad for colorblind people, and even for non-colorblind people it leads to incorrect interpretations of the data where the color luminance strongly varies).

This figure is now Figure 5 in the revised manuscript. We used the "gist_ncar" colormap in Figure 2 of our previous manuscript because that can depict a distinct boundary of the step-change in the mélange. In our revised manuscript, we have changed these colormaps to blue-yellow-red colormaps, which is more friendly to colorblind people.

We have tried many colormaps and this blue-yellow-red color scheme is our best choice if we consider both colorblind people and clarity of figures. For two reasons: 1) there are gaps in our

TRI data and we plotted them as white, so a blue-yellow-red can avoid mixture of high/low ends with void areas (shown in white); 2) a colormap changes from cold (blue) to hot (red) colors allows a wider range of varying pattern to be presented.

Figure 4: another change of colormaps. Red and green together in these maps is very hard to discern for colorblind people. It also gives the false impression of a step-change at the color change, even if the variable you are representing changes gradually. A perceptually uniform colormap would be much more appropriate to use.

This figure is now Figure 7 in the revised manuscript. We have changed the colormaps with blue-yellow-red scheme, that should have no problem with red-green colorblind people.

Color does not always change gradually because there is a step-change of elevation in the mélange, and an even larger step-change of elevation at the glacier front.

Figure 4e: pretty short timescale of observations... how does this fit into the longer term perspective? did you observe something usual or unusual? this is important to address to judge the implications of your findings (this is eventually made clear in the Supplementary Material, but only for the VERY astute reader, as it is buried rather deep...)

This figure is now Figure 7e in the revised manuscript. We have moved related information from the Supplementary Information to the main manuscript. See Figures 3 and 4 in our revised manuscript.

L 346: “bottom heights” refers to inferred bottom heights presumably? versus surface heights, which is what you measured.

Yes, “bottom heights” are inferred from measured surface heights based on hydrostatic equilibrium of the mélange. We have made changes to clarify this.

Supplementary Information: it would be helpful to have figures in line with the text, it is very tedious to scroll between the figures and the text describing them here. In the absence of line numbers, it is tedious to include comments on specific line numbers for further minor comments. Similar comments for figures (colormaps and captions) as for the main text.

We have moved several figures from the Supplementary Information to the main manuscript, and changed color maps and captions accordingly.

Responses to Reviewer #2:

Reviewer #2 (Remarks to the Author):

This manuscript described novel observations of the ice mélange in front of the calving terminus of Jakobshavn Isbrae, Greenland. The authors use a new radar-based approach to map time-

varying elevations near the mélange-glacier interface. The main conclusion is reiterated in the title: removal of tightly packed pro-glacial mélange results in rapid iceberg calving.

This conclusion is not supported by the measurements presented, and thus my recommendation is that the manuscript be rejected.

We think this reviewer may be confused by the mix of Eulerian versus Lagrangian reference frames used in the various figures, necessary to describe and interpret processes at a moving calving front over an extended period of time. We have clarified this in the paper; specific responses to his or her concerns given below.

There are two main problems with the paper. First, while step-change in mélange thickness shown in Figure 2b migrates somewhat towards the glacier terminus, the mélange thickness immediately in front of the terminus actually increases quite substantially. Figure 2e shows an increase of about 15 m at point E. Assuming the mélange is floating, this corresponds to a thickening of more than 100 m. Thus, if the back stress exerted by the mélange is indeed proportional to the thickness, this thickening would lead to a substantial increase in back stress and thus further reduction in iceberg calving. I fail to see how the much thinner mélange beyond the step change in elevation can overcome this back pressure.

We don't agree with this comment.

Figures 2c-2f show the elevation time series in a fixed reference frame relative to the Earth (Eulerian reference frame). The increase in Figure 2e is caused by thicker icebergs moved downstream to the location of previous thinner iceberg.

Ice thinning is calculated in a Lagrangian reference frame, which takes ice movement into account.

The second problem is more serious and seems to point at some problem with the data or data processing. The speed of Jakobshavn is around 50 m/day so over the three-week observation period, the terminus and mélange should have moved forward by 1 km or so. Yet the elevation maps in Figure 3 show no movement of the mélange. The three color maps shown in panels d-f are nearly identical, with only minor elevation changes. Similarly, the first movie provided in the SI shows no movement whatsoever of the glacier terminus region or of the thicker mélange in front of the terminus (except for three jerky adjustments of the whole region).

We don't agree with this comment.

The speed at the glacier terminus is ~37 m/d during the study period (Figures 6 and 8 in our revised manuscript). At this speed, the terminus moved forward by ~500 m. Figure 5 (previously Figure 3) clearly shows this movement, that the terminus and mélange moved towards the northwest, and the relative position between the selected box and the map frame is varying from Figures 5d-5f.

Movie S1 of the Supplementary Information also shows the advances of both the glacier and the mélange. The distance to map grid has changed throughout the observation period. We have noted in the caption that time in the movie runs nonlinearly (times are listed in movie legend).

I find it hard to believe that the calving rate of the glacier is zero throughout the period and jumps up on June 16 – when the observations have been terminated. If indeed the calving rate were zero from May 20 to June 19, the terminus should have advanced by 1.5 km. This advance is not shown in the radar observations – or I must be missing something.

Figure 3 in our revised manuscript (was in the SI in our previous submission) shows satellite and TRI images from May 20 to June 29. We suggest looking at these georeferenced images using the map grid or stationary points (e.g., rock) as references. For examples, using the grid of -2277 km on y axis, movement of the glacier front is obvious. Glacier advance can also be seen from Fig. 5a-5c that the glacier front (lower right) came into the map range of Fig.5c but not in Fig. 5a.

Responses to Reviewer #3:

This study uses a terrestrial radar interferometer to quantify changes in the mélange thickness extending in front of Jakobshavn Isbrae, Greenland. This mélange has been hypothesized to exert a buttressing force that can stabilize and inhibit calving from Jakobshavn and other marine terminating glaciers. Understanding the evolution of the pro-glacial mélange along with its mechanical interaction with the upstream glacier has thus been hypothesized to be an important piece of the ice dynamics puzzle. In this study, the authors are able to resolve the evolution of the height of mélange (or thickness, assuming freeboard) and to examine how this influences calving. Overall, this is a novel study that provides one of the first high resolution (temporal and space) examinations of a relatively newly discovered process. This type of data is likely to be of increasing interest to the glaciological community.

I did, however, have a few questions and comments. First off, I think one of the strong points of the paper is the attempt to examine different processes that control the evolution of mélange. To my understanding (and I apologize if I misunderstood), the authors try to parse out how much of the mélange thickness change is due to melting and how much is due to the divergence of mélange. The calculation is briefly described in the text and supplementary material, but it appears to assume that the mélange can be treated as an incompressible fluid. This seems like a questionable assumption given the blocky and granular nature of mélange described by the authors. Here, I think it would be helpful to more clearly state the assumptions of the calculations performed along with an honest assessment of any limitations.

We recognize the limitations of treating the mélange as an incompressible fluid, and we have stated the assumptions and possible limitations in our revised manuscript. The velocity (or displacement) fields derived from TRI data suggest that the mélange wedge had an extensional motion during the observation period, with positive divergence. Thus the incompressible assumption should have minor influence on the estimate of mean thinning rate. And the related error can be included in the uncertainty caused by ice density changes, which we allow to vary within $917 \pm_{-30}^{+5} \text{ kg m}^{-3}$.

Second, and perhaps related, the authors seem to assume that *mélange* can be treated as a fluid and thus it can form an ice shelf. If this is the case, why not try to fit an analytic ice shelf profile (e.g., Van der Veen *Fundamentals of Glacier Dynamics* p. 132, Equation 5.71) to see if the ice shelf really behaves like an ice shelf. The advantage of this approach is that the authors can infer the effective stiffness of ice associated with *mélange*. A distinct possibility is that the *mélange* does not behave like an ice shelf and instead the wedge is more like a compressional wedge. A good example of this is the wedge of sand that is formed in front of a bulldozer. In this case, the ice shelf is the bulldozer and it is ploughing through the *mélange* causing it to compress in front. The compression is the result of the motion of the calving front pushing *mélange* against pinning points and the thickness of the wedge in this case is the result of compression (often formed through jamming), but not the source of the compressive resistive stresses.

Although the detailed rheology of the *mélange* is beyond the scope of this study, we have fitted an analytical solution of ice shelf based on van der Veen (2013), assuming accumulation rate equals to mean melt rate. See the figure below, red and pink curve show a steady-state ice shelf profile in the along-flow direction, based on the analytical solution of van der Veen (2013) (p. 131-136), with “grounding line” located at the glacier front, exponent $n = 1$ and viscosity parameter $B = 5 \text{ kPa yr}^{1/3}$. The elevation profile is not well-fit with $n > 1.5$ or $B > 20 \text{ kPa yr}^{1/3}$, The viscosity parameter is much lower than a typical ice shelf.

However, we do not put this figure in the manuscript because we think this is not directly related to our conclusion and may cause over-extrapolation: the data we have is within a relatively small area, with highly dynamic environment, elevation changes a lot and the assumed “grounding line” (in really the *mélange* was floating during our observation period) position will also change parameters of the model in Van der Even (2013) significantly.

While we are not able to detect the formation of the thick *mélange* wedge, our data suggest that during the observation period, the *mélange* wedge had extensional motion.

Third, it is interesting that the thickness of the *mélange* is not correlated at all with the glacier velocity (or, presumably, the strain rate). Given the fact that the authors are attempting to estimate buttressing, it is straightforward to use that buttressing to estimate the effect it would have on the strain rate near the calving front (the authors cite several examples where this is done). This would give a sanity check on the magnitudes of buttressing estimated and ensure they are consistent with ice dynamics. Moreover, it is possible that the buttressing is not inhibiting the dynamics of the glacier (consistent with the lack of an effect on the velocity), but

instead is acting to suppress bergs that have already calved from detaching. Both are possible and teasing out which is actually happening would be an important advance.

We have explained possible reasons why the glacier velocity did not have significant response to thickness change or calving-like collapses of the mélange: The thick mélange wedge is tightly packed such that the back-stress decrease occurred at its down fjord side, but decayed rapidly with distance, so that the glacier front and mélange immediately in front of the glacier were not notably affected (Figures 8 c1, c2, d1, d2). Meanwhile, shear stress at the constricting margin increased. Once the shear stress reached to a threshold (yield stress), the remaining mélange wedge failed and the buttressing force from the mélange to the calving face dropped, resulting in glacier speed increase or iceberg calving. Previously, Podrasky et al. (2014) found that in a summer, response of glacier speed to tide variation decayed rapidly near the front, by a factor of e (~ 2.7) in ~ 2 km along flow line direction.

We are aware that the buttressing can inhibit detached icebergs from overturning. However, we are not currently able to distinguish that. Although TRI amplitude images and satellite images show wide surface crevasses, we are not able to detect the depth. Coherence of adjacent TRI scans is sensitive to relative displacements, and suggests shows that mélange downstream from the step-change has lower coherence than upstream, but there are no notable changes between the glacier and the mélange wedge. If there are blocks that were already detached from the glacier, their movements are very similar to the glacier front.

There is also some loose terminology in the paper that I found hard to understand. For example, what is a calving-like collapse and how is it different from a calving event? I think a little bit of explanatory text would help readers understand what the authors mean. A had similar questions about the step-step change “migrating” closer towards the calving front. To me, it seems as though the mélange is breaking up and the evacuation of mélange is causing the wedge to change shape.

We have explained a calving-like collapse event in our revised manuscript: “Except for some perturbations caused by calving-like collapse events within the mélange (mélange collapse in a way that is similar to iceberg calving at the glacier front, see Movie S1)”.

We have also explained the step-change migration by “During our two week observation period, the step-change in mélange elevation migrated towards the glacier via several calving-like collapse events, which progressively removed the downstream edge of the wedge (Movie S1 and S2)”.

Finally, I had significant problems interpreting the figures. Here, I would encourage the authors to adopt color schemes that are useful. There are numerous guides to assist this. For example, this webpage <https://knightlab.northwestern.edu/2016/07/18/three-tools-to-help-you-make-colorblind-friendly-graphics/> is quite helpful. Perceptually uniform color schemes are also a must. Figure 2, I suspect is an example where a better color scheme would make the figure informative (I see multiple low velocities, which I suspect are purely artifacts of the badly chosen color scheme). I highly recommend the authors use a tool (e.g., <http://www.colororacle.org>) to examine how their color schemes are likely to be perceived by

people with different eye conditions (color blindness is very common). I honestly see very little in the colored portions of Figures 2-4 and have largely ignored these figures. More color perceptive people may have a different reaction to these figures.

We have changed the color maps.

We tried all available color maps. However, due to large variation of elevation from downstream mélange to the glacier (~0 to ~100 m), it is challenging to emphasize the ~10 m elevation step-change of the mélange using a perceptually uniform color scheme, and at the same time be friendly for color blind people.

In our revision, we have used blue-yellow-red color scheme for all colormaps in the main manuscript. But we keep the color scheme for the supplementary Movie S2, which shows the ~10 m mélange elevation step-change distinctly from navy blue to green.

Detailed comments:

Abstract: Typically abstracts for Nature publications are fully referenced.

It is specified in the Nature Communications “guide to submission” that an abstract for an article should be unreferenced. See <https://www.nature.com/documents/ncomms-submission-guide.pdf>.

Page 2, line 19-20: I thought that surface melt now accounts for over half of the mass lost from Greenland?

We have changed this sentence to “Previous work () suggests that increasing ice discharge in marginal areas at or close to the glacier front is a major process contributing to recent ice loss in Greenland”.

Page 3 lines 43-44. It is misleading to argue that the effect of pro-glacial buttressing needs to be accounted for in models. Whether buttressing is important depends on the time and spatial scales considered—mélange is probably not of great importance for ice sheet of models of glacial-interglacial cycles. Moreover, most glacier models fail to accurately resolve terminus motion in the absence of mélange. In these cases, it is hard to see how “accurately quantifying” mélange buttressing would significantly improve model capability. A more limited, but accurate statement might be that understanding the observed evolution of Greenland outlet glaciers requires a better understanding of the role of mélange.

The sentence has been changes to “It has been suggested that the buttressing force on the glacier terminus depends on the thickness of the mélange (Burton et al., 2018; Krug et al, 2015). Therefore, to study the influence of mélange on calving, we analyzed time series of digital elevation models (DEMs), which allow us to monitor changes of mélange thickness.”

Page 3, line 53: Shouldn't error be plural? “We corrected height error*s*”

Done, changed to “errors”

Page 3, line 56: Shouldn't minutes be singular? "30 minute** window.

Done, changed to "30-minute"

Page 4, line 69: Are these calving events? Or do these take place in the mélange?

These took place in the mélange. We term them "calving-like collapse events", previously defined.

Page 4, line 75: Why does a lack of calving events indicate the mélange is tightly packed? There are *many* glaciers that calve less frequently than 17 days with little to no mélange. Why is this not a property of the glacier? The other lines of evidence are more convincing.

We have changed this paragraph. No significant motion in the nearby mélange caused by a small calving event is used to support a tightly packed mélange. And we have cited Amundson et al. (2010) that the entire lateral width of the mélange rapidly accelerated away from the glacier terminus even if only a small portion of the terminus fell into the mélange. There are two other lines of evidence: 1) Downstream mélange is easily disturbed by small calving-like collapses; 2) In another year at the same season, smaller calving events caused significant surrounding mélange motion.

Page 5, lines 104-105. The estimate of melting appears to assume that mélange is incompressible. This seems like a highly questionable assumption that needs to be either justified or, better yet, quantify the error associated with this assumption. See my previous comment.

Please see our previous responses.

Page 8, line 166, word "thickness" is repeated.

Removed the extra "of thickness"

Page 8, line 176. I'm assuming the "s" after 10 is means to indicate tens. I think this would be more clear if the authors just used a squiggly line to indicate approximately equal to.

Done.

Figures.

Figure 1. The orientation of panels a and b was hard for me to parse. It looks to be like Panel a is rotated relative to panel b, but this is hard to tell. It would be nice to see both figures oriented in the same direction (or with a North arrow in both to show how the images have been rotated).

There is no rotation between these two panels. Centerline of the TRI arc-scanning area does not point to the north, which may make some people feel these two panels have different orientations.

We have now added a North arrow in both panels. Note that these maps are all in polar stereographic projection with EPSG=3413. The X grid lines are not exactly parallel to south-north direction.

Figure 2a. It would be helpful to mark the calving front in Figure 2 a.

Done. See dashed red line in Figure 2a.

Figure 3: I don't understand what this figure is supposed to show. What is causing the strong variations in color?

This figure has become Figure 5 in our revised manuscript. This figure shows mélange thinning, one of the mechanisms for mélange mass loss. Direct measurement of mélange thinning near the glacier terminus is challenging and our observations provide a method to do that. Strong variations in color of Figures 5d-5f are due to variations in ice surface heights, reflecting highly variable iceberg sizes in the mélange. Variations in color of Figure 5g and 5h represent unevenly distributed thinning, such as thicker icebergs exposed larger surface areas to water and could have higher thinning rate.

References:

- Amundson J. M., Fahnestock M., Truffer M., Brown J., Lüthi M. P. & Motyka R. J. Ice mélange dynamics and implications for terminus stability, Jakobshavn Isbræ, Greenland. *J. Geophys. Res.* **115**, F01005 (2010).
- Burton, J. C., Amundson, J. M., Cassotto, R., Kuo, C. C. & Dennin, M. Quantifying flow and stress in ice mélange, the world's largest granular material. *Proc. Natl. Acad. Sci. USA*, **115**, 5105-5110 (2018).
- Khan S. A., Aschwanden A., Bjørk A. A., Wahr J., Kjeldsen K. K. & Kjær K. H. Greenland ice sheet mass balance: a review. *Rep. Prog. Phys.* **78**, 046801 (2015).
- Krug J., Durand G., Gagliardini O. & Weiss J. Modelling the impact of submarine frontal melting and ice mélange on glacier dynamics. *The Cryosphere* **9**, 989-1003 (2015).
- Podrasky D., Truffer M., Lüthi M., & Fahnestock M. Quantifying velocity response to ocean tides and calving near the terminus of Jakobshavn Isbræ, Greenland. *J. Glaciol.* **60**, 609-621 (2014).
- Van der Veen C. J. Fundamentals of glacier dynamics. CRC Press (2013).
- Xie S., Dixon T. H., Voytenko D., Deng F. & Holland D. M. Grounding line migration through the calving season at Jakobshavn Isbræ, Greenland, observed with terrestrial radar interferometry. *The Cryosphere* **12**, 1387-1400 (2018).

Reviewers' comments:

Reviewer #3 (Remarks to the Author):

This study shows TRI inferred digital elevation models for Jakobshavn Isbrae. This data shows a wedge of pro-glacial melange in front of the terminus. During the time period when the wedge is present, calving is suppressed. Moreover, calving resumes once the wedge is removed. This is an interesting result that shows some of the first detailed measurements of melange within a fjord, building on existing research that suggests melange can suppress calving. Overall, I think that the authors have done an excellent job addressing most of my concerns. I still have concerns with the perceptually non-uniform colormaps uses, but I will leave this to the authors and editor to determine.

I still have one more issue. It is unclear if the melange is actually buttressing the glacier or simply suppressing calving. If the melange is buttressing the glacier than it is exerting a back-force on the calving face of the glacier. There is a very simple test to determine if there is back-force: a back-force will reduce the longitudinal strain rate at the terminus. This is a direct consequence of applying the buttressing in a force-balance at the glacier terminus. The authors note that there is no change in speed of the glacier before and after the melange wedge dissipates. Speed should correlate with longitudinal strain rate, but may not be a sensitive indicator of changes in strain rate, especially if these are localized near the calving front. Here, I think the authors need to comment on why the data is insufficient to determine if the strain rate components do change. If the data clearly shows no change in strain rate, then it doesn't seem like the melange is actually buttressing the glacier. This would indicate that the melange is instead suppressing calving. This is possible if the melange doesn't reduce flow or prevent fracture of the glacier, but simply "holds" pre-existing through-cutting fractures from detaching as bergs. From the data presented, this second hypotheses seems equally plausible.

This also gets into the discussion starting around lines 362 that attempt to estimate the buttressing force/stress. I have to admit that I'm completely lost here. Part of the problem is that I'm not entirely sure how different terms are defined. The buttressing stress (actually, this should be a traction as stress is a tensor) has units of stress (kPa). What the authors call the buttressing force per unit thickness of the melange also has units of stress (kPa). But force has units of Newtons and Newtons/meter is not equal to a Pascal. Something here is off. It would be helpful to define exactly what the authors mean by buttressing stress vs a buttressing force.

Note also that a buttressing traction of ~240-480 kPa (line 327) will have a massive impact on the strain rate near the terminus. A stress near the upper end of the range will result in a ****compressive**** stress at the terminus and compressive longitudinal strain rates. This would surely be observable. But this may be due to my misunderstanding what the authors mean by contact-pressure? In contrast, a buttressing force of 10-20 kPa is small enough that it will have a negligible effect on the stress and strain rate. But a buttressing force this small is unlikely to be important in the dynamics of Jakobshavn because it is so much smaller than the mean state of stress. A 10 kPa stress is also smaller than the magnitude of the force associated with changing tidal elevations on the calving front.

Overall, I found the estimates of buttressing to be confusing, hard to follow and filled with assumptions. I understand what the authors are trying to do, but it might be helpful to streamline this section and/or move these calculations to supplementary material, where the authors have more space to elaborate.

With the exception of the discussion on buttressing, everything else in the manuscript read clearly and was well explained.

Reviewer #4 (Remarks to the Author):

I was asked to comment on the responses of the authors to the previous comment of Reviewer 2. In order to do that, I have read through the whole of the manuscript, the comments of R2, and the response of the authors to the comments of R2.

In general the response of the authors to the comments of R2 is to say that they don't agree with the comment and to then provide a reason for why they don't agree. Having read the paper in its entirety I think there are 2 main reasons why this situation has arisen. The first is that the manuscript is really not very well written. It is often hard to follow and, in responding to R2, the authors often have to rely on reference to evidence and arguments that are never presented clearly (if at all) in the manuscript. Hardly surprising then that R2 was confused or, more likely puzzled, by the text. The authors simply need to explain more clearly why they are saying what they are, and to make sure that they direct readers to figures that support the points made in the text - and this includes highlighting features in the figures that are critical to the explanation being proposed.

So, I would support R2's perspective that the paper needs more work and that this would mainly revolve around better integrating the figures into the manuscript to the point of highlighting much more strongly what are the key points a reader needs to take from each figure (and why they are important). I don't think it's acceptable for the authors to simply disagree with the reviewer's comments - from my perspective, they arise because of confusion generated by the way the manuscript is written, not because R2 is doing a poor job. Only the authors can fix that problem.

We are very grateful for the comments from the reviewers again. Their comments below are in black, our responses are in red.

Reviewer #3:

This study shows TRI inferred digital elevation models for Jakobshavn Isbrae. These data shows a wedge of pro-glacial mélange in front of the terminus. During the time period when the wedge is present, calving is suppressed. Moreover, calving resumes once the wedge is removed. This is an interesting result that shows some of the first detailed measurements of mélange within a fjord, building on existing research that suggests mélange can suppress calving. Overall, I think that the authors have done an excellent job addressing most of my concerns. I still have concerns with the perceptually non-uniform colormaps uses, but I will leave this to the authors and editor to determine.

Comments appreciated. We have changed color maps in the main paper and a perceptually uniform color scheme is used for sequential data.

I still have one more issue. It is unclear if the mélange is actually buttressing the glacier or simply suppressing calving. If the mélange is buttressing the glacier than it is exerting a back-force on the calving face of the glacier. There is a very simple test to determine if there is back-force: a back-force will reduce the longitudinal strain rate at the terminus. This is a direct consequence of applying the buttressing in a force-balance at the glacier terminus. The authors note that there is no change in speed of the glacier before and after the mélange wedge dissipates. Speed should correlate with longitudinal strain rate, but may not be a sensitive indicator of changes in strain rate, especially if these are localized near the calving front. Here, I think the authors need to comment on why the data is insufficient to determine if the strain rate components do change. If the data clearly shows no change in strain rate, then it doesn't seem like the mélange is actually buttressing the glacier. This would indicate that the mélange is instead suppressing calving. This is possible if the mélange doesn't reduce flow or prevent fracture of the glacier, but simply "holds" pre-existing through-cutting fractures from detaching as bergs. From the data presented, this second hypotheses seems equally plausible.

This is an interesting idea, and we have tried to test it with the available data. Unfortunately the answer is not clear cut, possibly because the strain rate estimates lack sufficient sensitivity, as detailed below.

With TRI measured line-of-sight (LOS) velocity maps, we can compute strain rate along the radar LOS. The absolute LOS strain rate may not be representative of longitudinal strain rate because of LOS geometry effect (e.g., the large blue area in Fig. S8c is due to LOS velocity gradient). For example, the glacier front moves approximately perpendicular to the radar LOS, thus LOS strain rate change here has limited sensitivity to actual longitudinal strain rate change. With velocities estimated by feature tracking of radar and satellite amplitude images, ice along-flow strain rate can also be estimated, but it is an average over a longer period (usually >1 day) compared to the LOS strain rate (2-minute).

We have analyzed strain rate changes along radar LOS and ice flow directions in our latest revision (Fig. 9, S7, S8). The data show no significant increase of strain rate for the majority of the glacier front, except for a small area close to the coast (lower right in Fig. 9d). Based on the available data, we can not preclude either of the two possibilities. When the elevation step-change is far away, the tightly-packed upstream mélange does not show increases in strain rate. Only when the calving-like collapses migrated the elevation step-change close enough to the glacier front does the thick mélange respond to reductions in buttressing force change associated with calving-like collapses. While the newly formed strain rate fissures in the mélange are located in areas where ice blocks already calved, they were not pre-existing features until the elevation step-change migrated sufficiently close. Similarly, even if there were detached ice blocks at the glacier front, strain rates across the not-yet collapsed ice blocks and the glacier front will not necessarily change if they were buttressed tightly. In summary, we cannot distinguish between the tightly-packed mélange and the glacier (and detached ice blocks, if present) using these strain rate data.

It is also worth noting that glacier responses to calving are highly sensitive to terminus position and flotation status. After major calving events within 1.5 days to the end of TRI observations, the glacier front retreated to a new position, and glacier front environment likely reconfigured. If there were any changes in speed or strain rate, they could also be caused by this reconfiguration (new terminus position, new flotation status, new buttressing condition). We therefore do not preclude the possibility of pre-existing through-cutting ice blocks.

This also gets into the discussion starting around lines 362 that attempt to estimate the buttressing force/stress. I have to admit that I'm completely lost here. Part of the problem is that I'm not entirely sure how different terms are defined. The buttressing stress (actually, this should be a traction as stress is a tensor) has units of stress (kPa). What the authors call the buttressing force per unit thickness of the mélange also has units of stress (kPa). But force has units of Newtons and Newtons/meter is not equal to a

Pascal. Something here is off. It would be helpful to define exactly what the authors mean by buttressing stress vs a buttressing force.

The reviewer is correct, buttressing stress is a tensor and has units of Pa, it equals the amount of force per unit area. In this paper, buttressing force is defined as “force per unit lateral-width” and it has a unit of N/m, following the definition used in Burton et al. (2018, PNAS) and Amundson et al. (2010, JGR). We have added a figure explaining this unit and the definition. It is a convenient unit for this problem, and represents the total force acting over the entire thickness of a 1-m lateral-segment of the glacier (the area between the two blue lines in Fig. S9a). We now specify the units whenever referring to stress or buttressing force.

Note also that a buttressing traction of ~240-480 kPa (line 327) will have a massive impact on the strain rate near the terminus. A stress near the upper end of the range will result in a **compressive** stress at the terminus and compressive longitudinal strain rates. This would surely be observable. But this may be due to my misunderstanding what the authors mean by contact-pressure? In contrast, a buttressing force of 10-20 kPa is small enough that it will have a negligible effect on the stress and strain rate. But a buttressing force this small is unlikely to be important in the dynamics of Jakobshavn because it is so much smaller than the mean state of stress. A 10 kPa stress is also smaller than the magnitude of the force associated with changing tidal elevations on the calving front.

The definitions have now been clarified in the revised manuscript, as described above. The actual mélange-glacier contact surface (shown by grain tiles on the glacier front of Fig. S9a) is where the mélange exerts buttressing force on the glacier, thus the buttressing stress at the contact surface should be larger than the equivalent buttressing stress on the entire calving surface estimated using a longitudinal coupling model based on speed increase associated with clearing of the mélange (Walter et al., 2012, *Annal Glaciol.*). Assuming mélange is homogeneous over its thickness, mélange buttressing stress on the contact can be estimated using the force relation between the mélange and glacier:

$$\sigma_g H_g = \sigma_m H_m$$

where σ_m is the buttressing stress on the mélange-glacier contact, σ_g is an estimated value of back-stress on the entire glacier front based on the longitudinal coupling model, H_m is mélange thickness, H_g is glacier thickness. The same relation has been also used in Todd & Christoffersen, 2014, TC.

A ~240-480 kPa pressure on the mélange-glacier contact (not the entire glacier front) is similar to water pressure for ice front at ~24-48 m below water. This pressure may change compressive longitudinal strain rates, but will be mostly on the mélange-glacier contact (near water line) and may not be reflected on the glacier surface (~100 m above water line).

To help understand this method, here we use water-glacier contact pressure as an example. For a H_w high water column, its pressure on the water-glacier contact is σ_w . And the equivalent pressure (σ_{gw}) on the entire glacier front (thickness is H_g) will have the relation:

$$\sigma_{gw}H_g = \sigma_wH_w$$

Because glacier front thickness and water depth here are usually close, σ_w and σ_{gw} are usually not distinguished. However, in our case, mélange thickness and glacier front thickness are significantly different.

A 10-20 kPa (these numbers became 11-22 kPa in the revision since we have discussed more of this, so we decided to use more accurate numbers) pressure change on the glacier front is a minimum estimate because the associated mean thickness change is a minimum estimate (it should decrease more within 1.5 days after the TRI campaign) and the wedge geometry was not considered in the calculation. This estimate is close to changes of stress associated with tidal changes. Our strain rate changes (based on the newly formed mélange fissures observed near the end of TRI observations) strongly suggest that more calving-like collapses would remove most of the remaining mélange wedge, and decrease the buttressing force by an additional amount (possibly reaching up to ~27-54 kPa if all remaining mélange wedge was moved away, see our revision). Previous studies show that even tidal changes could strongly affect glacier front motion at Jakobshavn (Xie et al., 2018, *The Cryosphere*). Without the extra buttressing force exerted from mélange, the glacier becomes more vulnerable to calving.

Overall, I found the estimates of buttressing to be confusing, hard to follow and filled with assumptions. I understand what the authors are trying to do, but it might be helpful to streamline this section and/or move these calculations to supplementary material, where the authors have more space to elaborate.

We agree that this discussion could have been clearer. We have now summarized the main points and calculations in the main paper, and put more of the details in the supplement, which allows an expanded discussion.

With the exception of the discussion on buttressing, everything else in the manuscript read clearly and was well explained.

Thank you very much!

Reviewer #4 (Remarks to the Author):

I was asked to comment on the responses of the authors to the previous comment of Reviewer 2. In order to do that, I have read through the whole of the manuscript, the comments of R2, and the response of the authors to the comments of R2.

In general the response of the authors to the comments of R2 is to say that they don't agree with the comment and to then provide a reason for why they don't agree. Having read the paper in its entirety I think there are 2 main reasons why this situation has arisen. The first is that the manuscript is really not very well written. It is often hard to follow and, in responding to R2, the authors often have to rely on reference to evidence and arguments that are never presented clearly (if at all) in the manuscript. Hardly surprising then that R2 was confused or, more likely puzzled, by the text. The authors simply need to explain more clearly why they are saying what they are, and to make sure that they direct readers to figures that support the points made in the text - and this includes highlighting features in the figures that are critical to the explanation being proposed.

So, I would support R2's perspective that the paper needs more work and that this would mainly revolve around better integrating the figures into the manuscript to the point of highlighting much more strongly what are the key points a reader needs to take from each figure (and why they are important). I don't think it's acceptable for the authors to simply disagree with the reviewer's comments - from my perspective, they arise because of confusion generated by the way the manuscript is written, not because R2 is doing a poor job. Only the authors can fix that problem.

Thank you for these comments and suggestions. We realize now that we did not explain some of the techniques well in the original manuscript. Since then we have rewritten a substantial portion of the manuscript, clarified the approach, and defined terminology whenever first introduced to minimize confusion. Insufficient clarification in the original manuscript gave rise to the first objection of reviewer 2.

Using one example that the previous reviewer R2 had highlighted: reviewer R2 thought Fig. 2e showed an increase of 15 m in surface height at point E, thinking that mélange became much thicker - which would contradict our interpretation. We have mentioned in the main paper and supplement multiple times that we estimate mélange ice loss in a Lagrangian reference frame, which is widely used to track individual fluid particles. This is because ice in the study area moves very fast (>30 m/d), hence an Eulerian reference frame fixed with respect to the solid Earth will cause significant interpretation problems, because an identical iceberg at different times will have significantly different locations relative to a fixed Earth. Therefore the heights of this hypothetical iceberg at different times do not corresponding to a fixed location relative to the solid Earth.

We have now specifically stated important definitions, terms, reference frames whenever used.

Refs:

Amundson J. M., Fahnestock M., Truffer M., Brown J., Lüthi M. P. & Motyka R. J. Ice mélange dynamics and implications for terminus stability, Jakobshavn Isbræ, Greenland. *J. Geophys. Res.* **115**, F01005 (2010).

Burton, J. C., Amundson, J. M., Cassotto, R., Kuo, C. C. & Dennin, M. Quantifying flow and stress in ice mélange, the world's largest granular material. *Proc. Natl. Acad. Sci. USA*, **115**, 5105-5110 (2018).

Todd J. & Christoffersen P. Are seasonal calving dynamics forced by buttressing from ice mélange or undercutting by melting? Outcomes from full-Stokes simulations of Store Glacier, West Greenland. *The Cryosphere* **8**, 2353-2365 (2014).

Walter J. I., Box J. E., Tulaczyk S., Brodsky E. E., Howat I. M., Ahn Y. & Brown A. Oceanic mechanical forcing of a marine-terminating Greenland glacier. *Ann. Glaciol.* **53**, 181-192 (2012).

Xie S., Dixon T. H., Voytenko D., Deng F. & Holland D. M. Grounding line migration through the calving season at Jakobshavn Isbræ, Greenland, observed with terrestrial radar interferometry. *The Cryosphere* **12**, 1387-1400 (2018).

Reviewers' comments:

Reviewer #3 (Remarks to the Author):

I believe that this is the third time I have reviewed this paper. The main point of the paper remains the same as the original submission: the authors document a wedge of melange in front of Jakobshavn and argue that the wedge of melange is tightly packed and suppressed iceberg calving. This is an interesting result that contributes to a growing body of knowledge. Overall, it is still unclear if the melange is suppression the detachment of icebergs or actually inhibiting fracturing of the calving front, but I think the authors make clear that the observations are indeterminate on this point. I don't have any additional comments or complaints that need to be addressed. Except, perhaps, for one question that might be philosophical in nature. I cannot easily tell the difference between melange and glacier ice from the velocities, elevations or strain rates. How do we know that the melange wedge is not actually still partially connected to the calving front and/or not just a heavily fractured piece of glacier ice? Overall, I think the authors have addressed my comments and I don't have anything to add, except for a couple of minor comments.

Abstract, line 21: Can you change the 10^7 N m^{-1} into a percent decrease or provide some context for the magnitude? My guess is that the typical readership will not have any context for this number.

Figure 9: I cannot tell the difference in this figure between the glacier and melange. Is it possible to draw the ice/melange boundary?

Reviewer #4 (Remarks to the Author):

This is a review of the revised version of the manuscript. I was previously asked to look at the initially submitted version to try to understand/explain a disagreement between the authors and one of the original reviewers. My sense was that the root of that disagreement lay in the quality of the writing of the paper, and I suggested that that needed some further attention before a decision could be made as to whether or not the paper could be considered to be publishable. Whilst I see some technical changes in the revised manuscript, I'm afraid that the underlying problems with the writing are largely unaddressed. Since I don't actually find the main conclusion of the paper - that the weakening and/or removal of proglacial melange in a fjord in front of a major tidewater glacier can result in calving of icebergs from that glacier - to be especially surprising (even though there is apparently one previous documented case where it did not happen), I find it hard to see the rationale for publication in a journal from either the Science or Nature groups of publications. It looks more like a Journal of Glaciology or The Cryosphere paper to me.

Detailed Comments (keyed by line number as in the submitted .pdf file).

27: than those from other glaciers

29: in addition to time-varying water levels?

30: has the potential to reduce uncertainty

36: reveal the details of ...

40: triggered by the incursion of warm ocean water...

41: The glacier's terminus is now embedded...

43: how stable the present terminus position will be in the longer term,

47: and that the buttressing force..... can be large enough to inhibit...

51: But presumably you also had to monitor the ice flow in the terminus region and the occurrence of iceberg calving in order to address this problem fully? Please make this clear.

60: than ice in the northern branch
63: for stationary rock areas
66-67: not clear whether this refers to errors in the DEM or in the data from the ground based radar interferometer (or both). Please clarify.
69: less large icebergs than what comparator?
70: comparing what with time series of data from stationary rock points?
82: I would say "tidally-induced changes in the surface elevation of the melange" - just to be clear
86: inferred how?
89: Movies S1 and S2
95: I think you mean that there was no calving from the main trunk of the glacier in the 17b days prior to the start of your field campaign. As written, the text implies that the entire trunk of the glacier is in the habit of calving. I presume this is not the case.
97: and it did not...
98-100: Ugly sentence that needs rewriting for clarity and English
101: did not result in motion of the surrounding melange
110: and that caused..
111: are those in which visible blocks calved, but...
135: advection of what?
136: driven by contact with...
137-138: Not clear what you did here to arrive at a melt thinning rate. This is addressed later but I think it needs to be spelled out before you present results. I'd also encourage a more focussed discussion of the errors and their likely sources/magnitudes relative to actual rates of change (I do appreciate that this is probably not easy given the study environment, but since this is a novel study I think readers would find it helpful to be presented with a clear statement of the full problem and how it was solved - not necessarily here - it could go in supplementary material or materials/methods).
141: exhibited insignificant changes
142: Start a new sentence at "Thus"
145: shape and location of what?
147: Does this include both surface and basal melting? Please clarify. If melange is fractured at surface and/or base, do you consider ice melt from fracture walls?
150: whereas the melange..
153: estimate the thinning rate
154: is separated (refers to pair, not images)
154: which correlation is being referred to here?
154: and the uncertainty
155: for 6-day periods
158: Don't abbreviate Figure when used in main body of text. OK when used as in line 151. Should be "Figures" here anyway as you are talking about more than a single figure element. Please treat this as a global comment as the same issue arises many times.
163: ice motion within the melange wedge
184: estimate for an area further downstream from Jakobshavn...
188: But I presume it would not be unreasonable for there to be inter-annual variability in these rates anyway as surface air temperatures, water temperatures etc. will all vary over time?
190: as well as modeling results, which show that...
191: driven principally (or maybe primarily would be better?)
194: Assuming that the..
205: when does the melange migrate towards the glacier front?
205: with a significant amount of...
206-208: While it moves downstream...it jumps upstream during each...collapse
210: estimate the glacier's calving rate
214-215: A 30 day period is a 30 day period! I think you mean "an unusually long period (30 days)"
217: reduced the buttressing force (decrease is not an active verb)

218: based on

219-220: Assuming that the average ice thickness at the glacier front is 800m, then the total ice mass calved over the 8.5 day period from June 20-29 2016, would be 6.7 +/- 0.8 Gt, nearly 3% of Greenland's annual mass loss (but clarify whether that refers to Greenland's annual mass in the study year or is calculated relative to some longer term mean value)

232: coincidence of...

239: change the buttressing force at the glacier front significantly

240-243: Not obvious to me what this sentence contributes to the story. As is, I would just delete it. If it is important, please make the point being made more obvious.

250: during the next..

251: If the buttressing force...

252: the shear stress

253: ..wedge must presumably have increased

256: moving the elevation step-change closer...

257: Can a fissure fail? Don't you mean that the ice failed in the vicinity of this pre-existing fissure?

260: of the buttressing...

260-261: calving events can occur at the glacier terminus.

261: Note that, within uncertainties, neither the line-of-sight nor the horizontal....changed significantly either before or after major calving events

264: "or other changes" is not very helpful. If you have specific things in mind then tell readers what they are. If you don't, then don't say anything.

266: more high strain rate zones than during the early...

267: is also adjacent to the area

268: after the end of the TRI observations.

269: Do you mean "the progressive formation of new fissures in the melange"?

277: within the upstream melange wedge

278: increase until

279: Ice surface velocity

282-283: ice thickness reaches a minimum immediately downstream...

286-287: represents the reduction in the buttressing force on the glacier front...

289: simplifying

294: Assuming the melange as homogeneous...This is clearly a pretty massive over-simplification, so it really needs some justification and discussion

317: in the speed...

318: we cannot determine whether part of the...

325: and their influence..

326: at daily and sub-daily timescales..

329: Past estimates of melange thickness used in modeling either relied on limited data (characterised by low spatial resolution and/or long revisit times) or assumed a uniform thickness for the melange

341: Supplementary Information

341: Maybe should just say "Estimates" - but I'm not sure if you are actually talking about how the estimation was done rather than what the estimates are. Needs clarification.

We are very grateful for the comments from the reviewers again. Their comments below are in black, our responses are in red.

Reviewer #3 (Remarks to the Author):

I believe that this is the third time I have reviewed this paper. The main point of the paper remains the same as the original submission: the authors document a wedge of melange in front of Jakobshavn and argue that the wedge of melange is tightly packed and suppressed iceberg calving. This is an interesting result that contributes to a growing body of knowledge. Overall, it is still unclear if the melange is suppression the detachment of icebergs or actually inhibiting fracturing of the calving front, but I think the authors make clear that the observations are indeterminate on this point. I don't have any additional comments or complaints that need to be addressed. Except, perhaps, for one question that might be philosophical in nature. I cannot easily tell the difference between melange and glacier ice from the velocities, elevations or strain rates. How do we know that the melange wedge is not actually still partially connected to the calving front and/or not just a heavily fractured piece of glacier ice? Overall, I think the authors have addressed my comments and I don't have anything to add, except for a couple of minor comments.

Thanks for the comments. We agree with the reviewer that it is not easy to distinguish melange and glacier ice based on just the velocities or strain rates. Presumably this reflects the fact that the mélangé is tightly packed. However, based on satellite images, the TRI amplitude images, and TRI-derived elevation maps, it is straightforward to distinguish melange and glacier because of large differences in surface roughness, albedo and other surface characteristics (e.g. Figure a, below). There is also a step change in

elevation. Evidence that the melange wedge is physically separate from the calving front includes:

- 1) Comparing Landsat-8 image acquired on 5 May 2016 and Sentinel-1 image acquired on 9 May 2016, we found no significant changes in the distribution pattern of icebergs immediately in front of the glacier (panels a and b of the figure below, major icebergs moved downstream but have fixed relative positions). Newly formed icebergs appeared on the Sentinel-1 image acquired on 21 May 2016, the ice cliff retreated and became more obvious, indicating a major calving event occurred between 9 May 2016 and 21 May 2016. We have added the two satellite images acquired on 5 May 2016 and 9 May 2016 to Figure 3 in our revision.

- 2) Figures 7a-7c give examples of elevation maps of the study area. There is a distinct increase in elevation between melange wedge and the glacier front (changes from purple to yellow in colormaps).

Abstract, line 21: Can you change the 10^7 N m^{-1} into a percent decrease or provide some context for the magnitude? My guess is that the typical readership will not have any context for this number.

We have added “similar to the range of buttressing force change due to tidal variation”.

Figure 9: I cannot tell the difference in this figure between the glacier and melange. Is it possible to draw the ice/melange boundary?

We have added yellow lines to show the boundary, please see revised Figure 9.

Reviewer #4 (Remarks to the Author):

This is a review of the revised version of the manuscript. I was previously asked to look at the initially submitted version to try to understand/explain a disagreement between the authors and one of the original reviewers. My sense was that the root of that disagreement lay in the quality of the writing of the paper, and I suggested that that needed some further attention before a decision could be made as to whether or not the paper could be considered to be publishable. Whilst I see some technical changes in the revised manuscript, I'm afraid that the underlying problems with the writing are largely unaddressed. Since I don't actually find the main conclusion of the paper - that the weakening and/or removal of proglacial melange in a fjord in front of a major tidewater glacier can result in calving of icebergs from that glacier - to be especially surprising (even though there is apparently one previous documented case where it did not happen), I find it hard to see the rationale for publication in a journal from either the Science or Nature groups of publications. It looks more like a Journal of Glaciology or The Cryosphere paper to me.

We believe this paper meets the aims and scope of Nature Communications based on several key points: 1) We have developed a method to estimate high precision elevation maps with terrestrial radar interferometry, making time-varying elevation monitoring over a large area possible for the first time, with vertical and spatial resolution that meets

or exceeds most satellite techniques, and extremely high time resolution, exceeding any previous technique (satellite or ground-based); 2) With these data, we present the first observations of a time-varying melange wedge in front of the fastest moving glacier in Greenland; 3) Observations from this study provide a quantitative understanding of the melange buttressing effect.

Detailed Comments (keyed by line number as in the submitted .pdf file).

27: than those from other glaciers

Added “those”.

29: in addition to time-varying water levels?

Added “time-varying water levels”.

30: has the potential to reduce uncertainty

Changed “can” to “has the potential to”

36: reveal the details of ...

Change “detailed” to “the details of”

40: triggered by the incursion of warm ocean water...

Added “the incursion of”

41: The glacier's terminus is now embedded...

Changed “Its terminus” to “The glacier’s terminus”

43: how stable the present terminus position will be in the longer term,

Changed “position is” to “terminus position will be in the longer term”

47: and that the buttressing force..... can be large enough to inhibit...

Changed “sufficiently large” to “large enough”

51: But presumably you also had to monitor the ice flow in the terminus region and the occurrence of iceberg calving in order to address this problem fully? Please make this clear.

We have added a sentence “Ice flow and glacier calving events were also analyzed based on TRI and satellite data (below and Xie et al.⁵)” Here.

60: than ice in the northern branch

Added “ice in”.

63: for stationary rock areas

Changed “on” to “for”.

66-67: not clear whether this refers to errors in the DEM or in the data from the ground based radar interferometer (or both). Please clarify.

Added “TRI” in from of “phase unwrapping errors”.

69: less large icebergs than what comparator?

Changed “less” to “few”.

70: comparing what with time series of data from stationary rock points?

Changed to “Accuracy and precision of TRI-derived DEM time series were assessed by computing root-mean-square deviation of time series for representative stationary rock points and slow-moving ice points, and by comparing with predicted tides.”

82: I would say "tidally-induced changes in the surface elevation of the melange" - just to be clear

Added “surface”.

86: inferred how?

Added “(based on TRI-derived surface elevations and the assumption of hydrostatic equilibrium)”.

89: Movies S1 and S2

Done.

95: I think you mean that there was no calving from the main trunk of the glacier in the 17b days prior to the start of your field campaign. As written, the text implies that the entire trunk of the glacier is in the habit of calving. I presume this is not the case.

In our revised manuscript, we have specified “the main trunk of the glacier” whenever referred to occurrence of calving.

97: and it did not...

Added “it”.

98-100: Ugly sentence that needs rewriting for clarity and English

We have changed this sentence to “In contrast, Amundson et al.¹⁰ investigated the interactions between Jakobshavn Isbræ and its proglacial mélange using year-round observations. They found that the entire lateral width of the mélange rapidly accelerated away from the glacier even when only a small portion of the terminus fell into the mélange.”

101: did not result in motion of the surrounding melange

Done.

110: and that caused..

Changed “causing” to “that caused”.

111: are those in which visible blocks calved, but...

Done.

135: advection of what?

Added “of mélange near the elevation step-changes”.

136: driven by contact with...

Added “contact with”.

137-138: Not clear what you did here to arrive at a melt thinning rate. This is addressed later but I think it needs to be spelled out before you present results. I'd also encourage a more focussed discussion of the errors and their likely sources/magnitudes relative to actual rates of change (I do appreciate that this is probably not easy given the study environment, but since this is a novel study I think readers would find it helpful to be presented with a clear statement of the full problem and how it was solved - not necessarily here - it could go in supplementary material or materials/methods).

Added “(more detail below and in the Supplementary Methods)”.

141: exhibited insignificant changes

Changed “has” to “exhibited”.

142: Start a new sentence at “Thus”

Done.

145: shape and location of what?

Added “of the selected Lagrangian area”.

147: Does this include both surface and basal melting? Please clarify. If melange is fractured at surface and/or base, do you consider ice melt from fracture walls?

We did not separate surface or basal melting. Through the entire manuscript, we only referred to “total thinning”, “divergent thinning”, “melt thinning” and never said we estimated surface or basal thinning. To clarify, we added “overall thickness decrease” to the sentence “We calculate melt thinning (overall thickness decrease) rate based on changes of surface elevation ...”.

Our thinning rate estimate refers to thinning in the vertical direction, as we were trying to analyze melange thickness changes. The selected area is tightly packed and no iceberg rollover occurred, ice melt from fracture walls will be included in overall thinning estimate (mean thinning over an area).

150: whereas the melange..

Added “the”.

153: estimate the thinning rate

Added “the”.

154: is separated (refers to pair, not images)

Changed “are” to “is”.

154: which correlation is being referred to here?

Added “in feature tracking”.

154: and the uncertainty

Added “the”.

155: for 6-day periods

Changed “in” to “for”.

158: Don't abbreviate Figure when used in main body of text. OK when used as in line 151. Should be "Figures" here anyway as you are talking about more than a single figure element. Please treat this as a global comment as the same issue arises many times.

We uses abbreviate Fig. following the format of other papers published in Nature Communications. We will follow the editor's advice on this.

163: ice motion within the melange wedge

Changed “at” to “within”.

184: estimate for an area further downstream from Jakobshavn...

Done.

188: But I presume it would not be unreasonable for there to be inter-annual variability in these rates anyway as surface air temperatures, water temperatures etc. will all vary over time?

Good point. We have added “Other factors, such as inter-annual variabilities in surface air temperatures or water temperatures can also cause the difference in ice melt rate estimates” to the end of this paragraph.

190: as well as modeling results, which show that...

Done.

191: driven principally (or maybe primarily would be better?)

Changed to “primarily”.

194: Assuming that the..

Added “that”.

205: when does the melange migrate towards the glacier front?

The whole sentence is “**The elevation step-change** in the melange migrates towards the glacier front”.

205: with a significant amount of...

Added “a”.

206-208: While it moves downstream...it jumps upstream during each...collapse

Done.

210: estimate the glacier's calving rate

Done.

214-215: A 30 day period is a 30 day period! I think you mean "an unusually long period (30 days)"

Done.

217: reduced the buttressing force (decrease is not an active verb)

Done.

218: based on

Done.

219-220: Assuming that the average ice thickness at the glacier front is 800m, then the total ice mass calved over the 8.5 day period from June 20-29 2016, would be 6.7 ± 0.8 Gt, nearly 3% of Greenland's annual mass loss (but clarify whether that refers to Greenland's annual mass in the study year or is calculated relative to some longer term mean value)

Done. Relative to the average annual mass loss between 2003 and 2014 from GRACE data.

232: coincidence of...

Changed “concurrency” to “coincidence”.

239: change the buttressing force at the glacier front significantly

Done.

240-243: Not obvious to me what this sentence contributes to the story. As is, I would just delete it. If it is important, please make the point being made more obvious.

Deleted.

250: during the next..

Added “the”.

251: If the buttressing force...

Added “the”.

252: the shear stress

Added “the”.

253: ...wedge must presumably have increased

Done.

256: moving the elevation step-change closer...

Changed “migrating” to “moving”.

257: Can a fissure fail? Don't you mean that the ice failed in the vicinity of this pre-existing fissure?

During the TRI observations period, no obvious fissures were observed on the glacier from the TRI velocity data. However, our TRI data coverage does not allow us to tell if there were fissures on the glacier immediately before large calving events.

The observed fissures in the melange wedge during TRI observation period can fail during calving-like events. E.g., the two large fissures in Figure 9c failed in the calving-like collapse shown in Figure 9d. Likewise, the newly formed fissure can fail in the next calving-like collapse after the TRI campaign.

260: of the buttressing...

Added “the”.

260-261: calving events can occur at the glacier terminus.

Done.

261: Note that, within uncertainties, neither the line-of-sight nor the horizontal....changed significantly either before or after major calving events

Done.

264: "or other changes" is not very helpful. If you have specific things in mind then tell readers what they are. If you don't, then don't say anything.

Deleted "or other changes".

266: more high strain rate zones than during the early...

Changed "compared to" to "than during".

267: is also adjacent to the area

Changed "neighboring" to "adjacent to".

268: after the end of the TRI observations.

Added "end of the".

269: Do you mean "the progressive formation of new fissures in the melange"?

Yes thank for the suggestion.

277: within the upstream melange wedge

Changed "with" to "within".

278: increase until

Changed “increases” to “increase”.

279: Ice surface velocity

Changed “Ice speed” to “Ice surface velocity”.

282-283: ice thickness reaches a minimum immediately downstream...

Done.

286-287: represents the reduction in the buttressing force on the glacier front...

Done.

289: simplifying

Changed “simply” to “simplifying”.

294: Assuming the melange as homogeneous...This is clearly a pretty massive over-simplification, so it really needs some justification and discussion

We deleted “is homogeneous and that”.

317: in the speed...

Added “the”.

318: we cannot determine whether part of the...

Changed “if” to “whether”.

325: and their influence..

Changed “its” to “their”.

326: at daily and sub-daily timescales..

Deleted “the”.

329: Past estimates of melange thickness used in modeling either relied on limited data (characterised by low spatial resolution and/or long revisit times) or assumed a uniform thickness for the melange

Done.

341: Supplementary Information

Done.

341: Maybe should just say "Estimates" - but I'm not sure if you are actually talking about how the estimation was done rather than what the estimates are. Needs clarification.

Changed to “Methods to estimate ...”

REVIEWERS' COMMENTS:

Reviewer #3 (Remarks to the Author):

I have reviewed this manuscript several times at this point. In general, I think the authors have adequately addressed my concerns. There are, of course, always nit-picky comments. But at this point, continuing to send the manuscript out to review is doing a disservice to the authors, and maybe the reviewers. I think the novelty of the methods warrants publication, but one of the other reviewers disagreed. The point has been reached where the editor needs to reach a decision instead of torturing the authors by continually sending out the manuscript for review. The fundamental content of the manuscript has not changed despite review by at least three reviewers, so there needs to be a decision rather than finding more reviewers or asking for additional changes.

We are very grateful for the comments from the reviewers.

Reviewer #3 (Remarks to the Author):

I have reviewed this manuscript several times at this point. In general, I think the authors have adequately addressed my concerns. There are, of course, always nit-picky comments. But at this point, continuing to send the manuscript out to review is doing a disservice to the authors, and maybe the reviewers. I think the novelty of the methods warrants publication, but one of the other reviewers disagreed. The point has been reached where the editor needs to reach a decision instead of torturing the authors by continually sending out the manuscript for review. The fundamental content of the manuscript has not changed despite review by at least three reviewers, so there needs to be a decision rather than finding more reviewers or asking for additional changes.

Thank you!